# Reduced Order Models for the Quasi-Geostrophic Equations: A Brief Survey



**Changhong Mou [1], Zhu Wang [2], David R. Wells [3], Xuping Xie [4] and Traian Iliescu [1,\*]**

[1] Department of Mathematics, Virginia Tech, Blacksburg, VA 24061, USA; cmou@vt.edu
[2] Department of Mathematics, University of South Carolina, Columbia, SC 29208, USA; wangzhu@math.sc.edu
[3] Department of Mathematics, University of North Carolina, Chapel Hill, NC 27516, USA; drwells@email.unc.edu
[4] Courant Institute of Mathematical Sciences, New York University, New York, NY 10012, USA; xxie@nyu.edu
[\*] Correspondence: iliescu@vt.edu

**Abstract:** Reduced order models (ROMs) are computational models whose dimension is significantly lower than those obtained through classical numerical discretizations (e.g., finite element, finite difference, finite volume, or spectral methods). Thus, ROMs have been used to accelerate numerical simulations of many query problems, e.g., uncertainty quantification, control, and shape optimization. Projection-based ROMs have been particularly successful in the numerical simulation of fluid flows. In this brief survey, we summarize some recent ROM developments for the quasi-geostrophic equations (QGE) (also known as the barotropic vorticity equations), which are a simplified model for geophysical flows in which rotation plays a central role, such as wind-driven ocean circulation in mid-latitude ocean basins. Since the QGE represent a practical compromise between efficient numerical simulations of ocean flows and accurate representations of large scale ocean dynamics, these equations have often been used in the testing of new numerical methods for ocean flows. ROMs have also been tested on the QGE for various settings in order to understand their potential in efficient numerical simulations of ocean flows. In this paper, we survey the ROMs developed for the QGE in order to understand their potential in efficient numerical simulations of more complex ocean flows: We explain how classical numerical methods for the QGE are used to generate the ROM basis functions, we outline the main steps in the construction of projection-based ROMs (with a particular focus on the under-resolved regime, when the closure problem needs to be addressed), we illustrate the ROMs in the numerical simulation of the QGE for various settings, and we present several potential future research avenues in the ROM exploration of the QGE and more complex models of geophysical flows.

**Keywords:** reduced order models; quasi-geostrophic equations; closure models

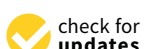



## 1. Introduction

### 1.1. Reduced Order Models (ROMs)

Reduced order modeling aims at answering the following question:

$$\boxed{\text{For a given system, what is the model with the minimum number of degrees of freedom?}} \tag{1}$$

The resulting models, called reduced order models (ROMs), can decrease the computational cost of traditional full order models (FOMs) (i.e., models obtained through classical numerical discretizations, such as finite element, finite difference, finite volume, or spectral methods) by orders of magnitude without a significant decrease in numerical accuracy. Thus, ROMs can be used in the efficient numerical simulation of problems that require numerous runs, e.g., uncertainty quantification, control, and shape optimization.

ROMs come in different flavors. Projection ROMs have been used in the numerical simulation of both nonlinear [1–3] and linear [4] systems. In particular, projection ROMs

have been successful in the numerical simulation of complex fluid flows [2,5–7]. In this survey, we exclusively consider projection ROMs that answer question (1) as follows:

> To construct the ROM, use numerical or experimental data to find the "best" basis.    (2)

Once the "best" basis is found, the ROM is constructed by using projection methods. In Galerkin projection ROMs, the trial and test spaces are the same; in Petrov-Galerkin projection ROMs, the trial and test spaces are different. In this paper, we focus on Galerkin projection ROMs.

Specifically, to approximate the dynamics of a flow variable $u$ of a given system

$$\dot{u} = f(u), \tag{3}$$

the ROM strategy proceeds as follows:

---

**Algorithm 1** ROM Strategy

---

1: Use numerical or experimental data to choose modes $\{\varphi_1, \ldots, \varphi_R\}$, which represent the recurrent spatial structures in the flow.

2: Choose the dominant modes $\{\varphi_1, \ldots, \varphi_r\}$, $r \leq R$, as basis functions for the ROM.

3: Use a Galerkin truncation $u_r(x, t) = \sum_{j=1}^{r} a_j(t)\varphi_j(x)$.

4: Replace $u$ with $u_r$ in (3).

5: Use a Galerkin projection of the PDE obtained in step (4) onto the ROM space $\mathbf{X}^r := \text{span}\{\varphi_1, \ldots, \varphi_r\}$ to obtain the ROM:

$$\dot{a} = F(a), \tag{4}$$

where $a(t) = (a_i(t))_{i=1,\ldots,r}$ is the vector of coefficients in the Galerkin truncation in step (3) and $F$ comprises the ROM operators.

6: In an offline stage, compute the ROM operators (e.g., vectors, matrices, and tensors), which are preassembled from the ROM basis.

7: In an online stage, repeatedly use the ROM (4) for various parameter settings and/or longer time intervals.

---

At this point, several remarks are in place.

**ROMs are Galerkin methods with a data-driven basis**: First, we note that the general form of projection ROMs (outlined in Algorithm 1) is strikingly similar to the general form of classical Galerkin methods used in a finite element, spectral, or spectral element context. Conceptually, the main difference between ROMs and classical Galerkin discretizations is the way the basis is constructed: In classical Galerkin methods, the basis is universal, i.e., it is the same for all the problems. For example, for finite elements, the basis functions are piecewise polynomial functions on a given mesh. In projection ROMs, however, the basis is a *data-driven basis*, i.e., a basis constructed from problem data. Thus, the ROM basis is adapted to the specific problem (see steps (1)–(2) in Algorithm 1): Once the problem changes, the ROM basis changes accordingly.

While the choice of basis is the main conceptual difference between ROMs and classical Galerkin methods, this choice can make a tremendous difference in the computational cost: For example, for a two-dimensional flow past a circular cylinder at a Reynolds number Re = 1000, a finite element discretization requires $\mathcal{O}(10^5)$ degrees of freedom, whereas a

ROM requires $\mathcal{O}(10)$ degrees of freedom [8,9]. Thus, for this particular test case, *the ROM dimension is four orders of magnitude lower than the FOM dimension*.

**Recurrent, Dominant, Coherent Spatial Structures**: ROMs do not work well for all problems. ROMs are numerical methods and, like any other numerical method, ROMs work well for certain classes of problems and not so well for other classes of problems. One class of problems for which ROMs have been particularly successful is flows that display *recurrent, dominant, coherent structures*. A classical example in this class is the two-dimensional flow past a circular cylinder, which has become the workhorse of ROMs for fluid flows [5,6,9]. The flow past a circular cylinder displays coherent spatial structures (the von Karman vortex street) that continuously recur in time. One can show that a few such structures have significantly higher kinetic energy content than the remaining structures, and, therefore, are expected to dominate the dynamics of the underlying system. Indeed, as mentioned above, for the two-dimensional flow past a circular cylinder, the dimension of the ROM constructed with these dominating structures can be four orders of magnitude lower than the FOM dimension. Thus, for this test problem, ROMs work extremely well. Other problems that display recurrent, dominant, coherent structures, for which ROMs work well, include: (i) lid driven cavity flow [10]; (ii) flow past a backward facing step [11]; (iii) flow in a constrained channel [12,13]; and (iv) flow in the boundary layer of a pipe [2].

We emphasize, however, that there are classes of problems for which ROMs do not work well. Homogeneous flows are one such example. Indeed, for homogeneous flows, it was proved in [2,14] that one of the most popular ROM techniques yields a ROM basis that is identical to the Fourier basis. Thus, in this case, the resulting ROM is nothing but a spectral method, which does not reduce the FOM dimension.

The take-home message is that *ROMs are appropriate for problems that display recurrent, coherent, dominant spatial structures. However, for problems that do not display these types of spatial structures (e.g., homogeneous flows), ROMs are not appropriate since they cannot reduce the FOM computational cost.*

### 1.2. ROMs for the Quasi-Geostrophic Equations

*ROMs are an excellent fit for the numerical investigation of ocean flows.* Indeed, large-scale ocean circulation includes large-scale coherent structures (gyres) that recur in time and permanent gyres (e.g., the Sargasso Sea) that have a relatively high kinetic energy content. Thus, as pointed out above, ROMs could enable an efficient and relatively accurate numerical simulation of large scale ocean circulation, decreasing the FOM computational cost by orders of magnitude and making possible efficient ensemble calculation and uncertainty quantification for climate modeling and weather prediction.

However, generating FOM data to build the ROM basis can be a daunting task. Specifically, using an accurate mathematical model (e.g., the Boussinesq equations), including all the relevant flow variables, and using realistic parameters, could require enormous computational resources on state-of-the-art computational platforms, both in terms of CPU time and memory. Thus, various *simplified mathematical models* for the large scale ocean circulation have been proposed over the years [15–17]. These simplified models are constructed by using *asymptotic expansions* with respect to both the time scales and the length scales. The *rotation* and *stratification* are the two main effects that are used to construct simplified models for geophysical flows.

One of the most popular simplified models for large scale ocean circulation is the *quasi-geostrophic equations (QGE)* (also known as the barotropic vorticity equations), which were proposed in the late 1940s by Jule Charney [18]. The QGE are a simplified model for geophysical flows in which rotation plays a central role, such as wind-driven ocean circulation in mid-latitude ocean basins. Specifically, the QGE ensure a near-geostrophic balance, i.e., the pressure gradient almost balances the Coriolis force (which is due to rotation). The one-layer QGE do not include stratification effects, but the $N$-layer or continuously stratified QGE model stratification.

The computational cost of numerical simulations of large scale ocean flows is significantly lower for the QGE than for the full-fledged Boussinesq equations. Since the QGE represent a practical compromise between the efficient numerical simulations of ocean flows and the accurate representation of large scale ocean dynamics, these equations have been often used in the testing of new numerical methods for ocean flows. Thus, to understand the potential of using ROMs for the efficient numerical simulation of ocean flows, ROMs have been tested on the QGE for various parameter settings. Of course, once the ROMs are calibrated for the simplified (yet relevant) setting of the QGE, they should be extended to more realistic mathematical models, such as the Boussinesq equations. In this brief survey, we summarize some of the ROM developments for the QGE.

The rest of the paper is organized as follows: In Section 2, we present and discuss the QGE. In Section 3, we summarize the main types of numerical discretizations used to generate the FOM data for the ROM construction. In Section 4, we present the Galerkin ROM approach for the QGE. In Section 5, we illustrate numerically the QGE reduced order modeling for one test case. Finally, in Section 6, we present conclusions and outline open problems in the reduced order modeling of the QGE.

## 2. Quasi-Geostrophic Equations (QGE)

The QGE describe the motion of stratified, rotating flows, and have been used extensively for modeling mid-latitude oceanic and atmospheric circulations. In 1950, a single-layer quasi-geostrophic model was used for modeling the atmospheric dynamics in the first successful numerical weather prediction performed on the ENIAC digital computer [18], which led to "enormous scientific advance", in Richardson's words [15,19,20]. Since then, the QGE have been widely investigated and applied in weather prediction and climate modeling.

The QGE can be derived from the primitive equations, that is, the incompressible Navier–Stokes equations under the Boussinesq approximation in a rotating framework [21–24]. The equations in Cartesian coordinates on a plane $\Omega$ tangent to the sphere read:

$$\frac{Du}{Dt} - f_c v = -\frac{1}{\rho}\frac{\partial p}{\partial x} + \frac{\partial}{\partial x}\left(\mathcal{A}\frac{\partial u}{\partial x}\right) + \frac{\partial}{\partial y}\left(\mathcal{A}\frac{\partial u}{\partial y}\right) + \frac{\partial}{\partial z}\left(\nu_E\frac{\partial u}{\partial z}\right), \tag{5}$$

$$\frac{Dv}{Dt} + f_c u = -\frac{1}{\rho}\frac{\partial p}{\partial y} + \frac{\partial}{\partial x}\left(\mathcal{A}\frac{\partial v}{\partial x}\right) + \frac{\partial}{\partial y}\left(\mathcal{A}\frac{\partial v}{\partial y}\right) + \frac{\partial}{\partial z}\left(\nu_E\frac{\partial v}{\partial z}\right), \tag{6}$$

$$0 = -\frac{1}{\rho}\frac{\partial p}{\partial z} - g, \tag{7}$$

$$0 = \frac{\partial u}{\partial x} + \frac{\partial v}{\partial y} + \frac{\partial w}{\partial z}, \tag{8}$$

$$\frac{D\rho}{Dt} = \frac{\partial}{\partial x}\left(\mathcal{A}\frac{\partial \rho}{\partial x}\right) + \frac{\partial}{\partial y}\left(\mathcal{A}\frac{\partial \rho}{\partial y}\right) + \frac{\partial}{\partial z}\left(\kappa_E\frac{\partial \rho}{\partial z}\right), \tag{9}$$

where $u$, $v$, and $w$ are velocity components in the $x$, $y$, and $z$ directions, $\frac{D}{Dt} = \frac{\partial}{\partial t} + u\frac{\partial}{\partial x} + v\frac{\partial}{\partial y} + w\frac{\partial}{\partial z}$ is the material derivative, $\rho$ is density, $p$ is the pressure, $f_c$ is the Coriolis force, and eddy viscosity and diffusivity coefficients $\mathcal{A}$, $\nu_E$, and $\kappa_E$ are either constant or functions of flow variables and grid parameters. The dimensionless Rossby number Ro is defined as Ro $= \frac{U}{f_c L}$, in which $U$ and $L$ represent the velocity and length scale of the geophysical flows. The Rossby number essentially characterizes the strength of inertia compared to the Coriolis and pressure forces. Another dimensionless number is the Ekman number, which is defined as Ek $= \frac{\nu_E}{\Omega H^2}$, with $H$ the vertical extent of the flow. Since Ro is the ratio of the respective scales $\frac{U^2}{L}$ and $f_c U$ of the first two terms in (5) and (6), and since Ek measures the ratio of viscous forces to Coriolis forces, when both *Ro* and *Ek* are much smaller than 1 (e.g., *Ro* = 0.0036 in Section 5), the Coriolis term dominates the left hand sides of momentum

Equations (5) and (6), and the equations can be simplified, yielding the geostrophic balance:

$$-f_c v = -\frac{1}{\rho}\frac{\partial p}{\partial x}, \tag{10}$$

$$f_c u = -\frac{1}{\rho}\frac{\partial p}{\partial y}. \tag{11}$$

The resulting system reaches an equilibrium state in which the pressure gradient balances perfectly with the Coriolis force. When the Rossby and Ekman numbers are still small, but not nearly zero, the flow only achieves a near-geostrophic balance. Considering the beta-plane approximation $f_c = f_0 + \beta y$ and ignoring the stratification effect, one can obtain the single layer QGE by regular perturbation analysis [17,25,26]. The resulting equations are usually put in the following streamfunction-potential vorticity two-dimensional formulation:

$$\frac{\partial q}{\partial t} + J(q, \psi) = \mathrm{Re}^{-1}\Delta q + F_e, \tag{12a}$$

$$q = -\mathrm{Ro}\,\Delta\psi + y, \tag{12b}$$

where $\mathrm{Ro} = \frac{U}{\beta L^2}$ is the redefined Rossby number [27–29], $\mathrm{Re} = \frac{UL}{A}$ is the Reynolds number, $\psi$ is the streamfunction, $q$ is the potential vorticity, $J(q, \psi) = \frac{\partial q}{\partial x}\frac{\partial \psi}{\partial y} - \frac{\partial q}{\partial y}\frac{\partial \psi}{\partial x}$ is the Jacobian, $F_e$ is the external forcing, and $\beta y$ measures the beta-plane effect from the Coriolis force due to rotation. By eliminating the potential vorticity, we can obtain the pure streamfunction formulation:

$$-\frac{\partial}{\partial t}(\Delta\psi) + \mathrm{Re}^{-1}\Delta^2\psi - J(\Delta\psi, \psi) - \mathrm{Ro}^{-1}\frac{\partial\psi}{\partial x} = \mathrm{Ro}^{-1}F_e. \tag{13}$$

Equations (12) and (13) are supplemented by boundary conditions, such as $\psi = \frac{\partial\psi}{\partial n} = 0$ on $\partial\Omega$. More details regarding the parameters and nondimensionalization of the QGE are given in, e.g., [8,30–33]. Note that the velocity can be recovered from the streamfunction according to the following formula:

$$v = \left(\frac{\partial\psi}{\partial y}, -\frac{\partial\psi}{\partial x}\right). \tag{14}$$

One can also introduce the vorticity $\omega = \mathrm{Ro}^{-1}(q - y)$ and recast the QGE (12) in the following streamfunction-vorticity formulation:

$$\frac{\partial\omega}{\partial t} + J(\omega, \psi) - \mathrm{Ro}^{-1}\frac{\partial\psi}{\partial x} = Re^{-1}\,\Delta\omega + \mathrm{Ro}^{-1}F_e, \tag{15a}$$

$$\omega = -\Delta\psi. \tag{15b}$$

This form is close to the streamfunction-vorticity formulation of the two-dimensional Navier–Stokes equations, but it has an additional convection term $\mathrm{Ro}^{-1}\frac{\partial\psi}{\partial x_1}$ and the forcing term is scaled by $\mathrm{Ro}^{-1}$ due to the rotation effect of the Earth. Such rotation effect can significantly change the behavior of QGE and yields a strong boundary layer in the solution, as shown in Figure 1: When Ro is unphysically large (i.e., close to 1) we have larger, circular gyres with lower kinetic energy but when Ro is decreased the gyres both increase in energy (due to increased forcing), which can be seen in the higher vorticity magnitudes, and move westward (due to the convection term).

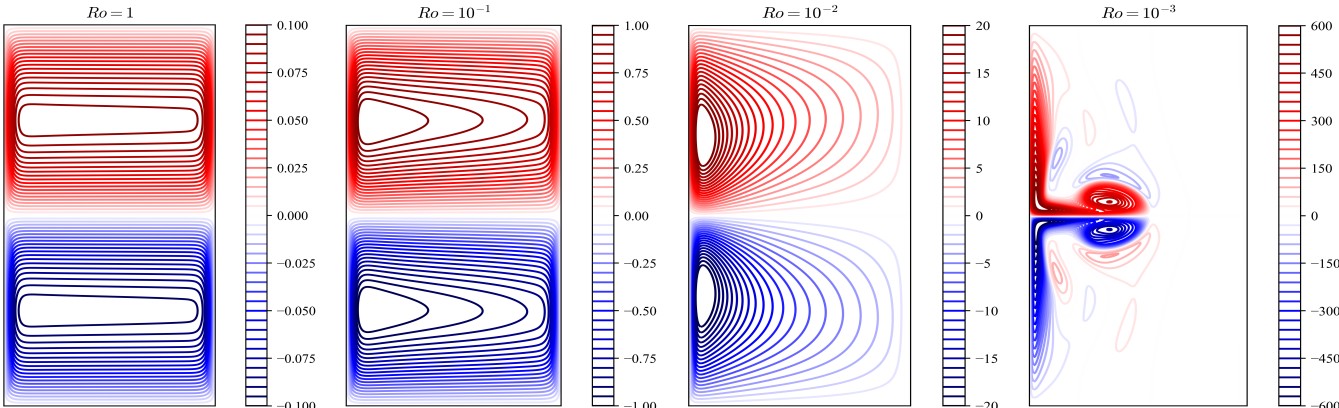

**Figure 1.** Solutions (vorticities) of the QGE subject to different Rossby numbers on the rectangular domain $[0,1] \times [0,2]$ when Re $= 100$ and $F_e = \sin(\pi(y-1))$ at $t = 0.1$. From left to right: Ro $= 1, 0.1, 0.01,$ and $0.001$. It is seen that decreasing the Rossby number yields a sharper western boundary layer.

When the fluid of interest is homogeneous, that is, no stratification is considered, we have the single layer QGE model. This is what we mainly focus on in this paper. However, to better approximate a continuously stratified fluid, a multi-layer model can be developed that assumes that the fluid consists of stacked isopycnal layers, the variation in the thickness of each layer is small compared to its mean thickness, and adjacent layer equations are coupled through the quasi-geostrophic potential vorticity [17]. Numerical investigations of multi-layer QGE have been made, for instance, in [34–36].

## 3. Full Order Model (FOM)

To generate FOM numerical data to construct the ROM basis, the QGE (12) need to be discretized both in space and in time. Popular QGE spatial discretizations include finite difference (FDM), finite volume (FVM), finite element (FEM), and pseudospectral methods. Essentially all discretizations use the method of lines (e.g., Runge-Kutta methods or other standard ODE solvers) to discretize in time. In this section, we survey each of these spatial discretizations for the QGE and, where available, comment on the existing numerical analysis results.

The primary intent of this paper is to survey the state-of-the-art for reduced order modeling of the QGE. While FOMs are required by ROMs, this section is not intended to be an exhaustive survey of the literature on the subject and instead only highlights the major trends.

### 3.1. Finite Difference Methods for the QGE

It is straightforward to apply the FDM to a geophysical flow model on rectangular grids. These were the first methods used [18,37] to simulate geophysical flows. In particular, the Arakawa grids were introduced by Arakawa and Lamb [38] to conserve energy and enstrophy at the grid level by effectively locating state variables across the mesh (i.e., a staggered-grid representation instead of nodal or cell-centered). See, e.g., [15,39] for detailed discussions. Among this class of grids, the C-grid places scalar quantities at the cell centers, while specifying the normal velocity components at the cell edges (which is essentially the classic MAC scheme [40]). Because of its excellent representation of the inertial-gravity waves, it has been widely used in geophysical flow simulations, for instance, for solving QGE in [33] and is the standard solver in the Modular Ocean Model version 6 [41]. Staggered-grid grid approximations like the C-grid can be thought of as either finite difference or finite volume schemes since the various velocity fluxes are explicitly solved for at cell faces rather than being reconstructed first from cell-centered values—this is the fundamental property that gives, e.g., the MAC scheme exactly zero divergence at cell centers (when calculated with standard second-order difference operators). Such schemes can also be extended to work with various turbulence modelling strategies [33,42–44].

### 3.2. Finite Volume Methods for the QGE

Like the staggered-grid finite difference schemes, the principal advantage of the FVM is preservation of the essential conservative quantities for the governing equations of geophysical fluid flows while additionally dealing with unstructured grids (i.e., complex geometries) more easily. This avoids the need for discretizing boundaries with staircasing, which results in inaccurate modelling of coastal phenomena like Kelvin waves [45]). This combination of properties makes the FVM the most common method for large-scale ocean simulations, such as those performed with [41] or [46].

Methods that use C-grid like discretizations (i.e., storing normal velocities on cell faces and mass or pressure in cell centers) on arbitrarily structured meshes must additionally introduce corrective measures to deal with the reconstruction of the tangential velocity (which is required by the discretization of the Coriolis force) [47–49]. These generalized C-grid methods are applicable to a wide class of meshes including latitude-longitude grids, Delaunay triangulations, Centroidal Voronoi tessellation (CVT), and spherical CVT. A different approach to overcome this issue was considered in [50], where the non-staggered Z-grid scheme [51] was used for the QGE model.

### 3.3. Pseudospectral and Spectral Methods for the QGE

Like finite difference methods, pseudospectral methods (due to their immediate applicability to hypercube geometries) have been used in a variety of different ways in QGE solvers. Some QGE solvers, like the one used in [52], use a pseudospectral discretization to compute turbulence statistics. Alternatively, some finite difference methods use a pseudospectral interpretation of solution grid values to do fast Laplace solves with a multidimensional discrete sine transformation [32,53] or resolve stability problems from nonlinearities via dealiasing [54,55].

Furthermore, pseudospectral methods have been used for the spatial discretization of the QGE [56,57]. The FOM results used in this paper to construct the ROM basis in Section 5 were also generated with a pseudospectral method. By *pseudospectral* we mean that spatial derivatives in (12) are evaluated by performing a discrete sine transform, wave number multiplication, and an another discrete sine transform in which dealiasing (the 3/2 s rule from [58]) is used in the nonlinear term of (12) for stability. The FOM solver exploits the homogeneous boundary conditions to ignore even-numbered Fourier modes (i.e., the `RODFT00` transformation in [59]). Since this method permits very fast evaluation of spatial derivatives we essentially treat them as a black box operation as part of an explicit ODE solver for evolving the Fourier coefficients in time. The largest stable timestep is found by using the power method for computing the principal eigenvalue of the linearized discretization of (12). Numerical experiments imply that setting a strict error tolerance on the error caused by the ODE solver requires smaller timesteps than the one required for ODE stability, which validates the choice of explicit methods for the relevant range of Reynolds numbers and grid resolutions. See Section 5.4 for additional details on the numerical experiments used in this manuscript.

### 3.4. Finite Element Methods for the QGE

The FEM is particularly appealing because it combines advantages of multiple methods. It can easily handle adaptive mesh refinement and complex geometries (like the FVMs), but also can create higher-order schemes (like pseudospectral methods) at the same time, like the discretization used in [30,60], which is shown in Figure 2 (see also [61–63]). The first FE approximation of the QGE, to the best of our knowledge, was a scheme based on the mixed formulation developed in [64]. The conservation properties and stability of the FE discretization were proved as well as the suboptimal convergence of the FE method. The performance of FEM on simulating a multilayer QGE of ocean circulations has been compared to the FDM in [65]. Since the vorticity-streamfunction formulation (15) of the QGE results in a second-order PDE, for a conforming finite element discretization, a $C^0$ element can be utilized. Considering the finite element spaces $W_\psi \subset H_0^1(\Omega) \bigcap W^{1,4}(\Omega)$,

$W_\omega \subset H^1(\Omega)$ (see, e.g., [66] for the definition of these finite element spaces), the finite element discretization reads: Find $\psi_h \in W_\psi$ and $\omega_h \in W_\omega$ satisfying

$$
\begin{cases}
\left(\frac{\partial \omega_h}{\partial t}, \phi_h\right) + (J(\omega_h, \psi_h), \phi_h) = -\mathrm{Re}^{-1}(\nabla \omega_h, \nabla \phi_h) + \mathrm{Ro}^{-1}\left(\frac{\partial \psi_h}{\partial x}, \phi_h\right) + \mathrm{Ro}^{-1}(F_e, \phi_h) & \forall \phi_h \in W_\psi \\
(\omega_h, v_h) = (\nabla \psi_h, \nabla v_h) & \forall v_h \in W_\omega
\end{cases}
\tag{16}
$$

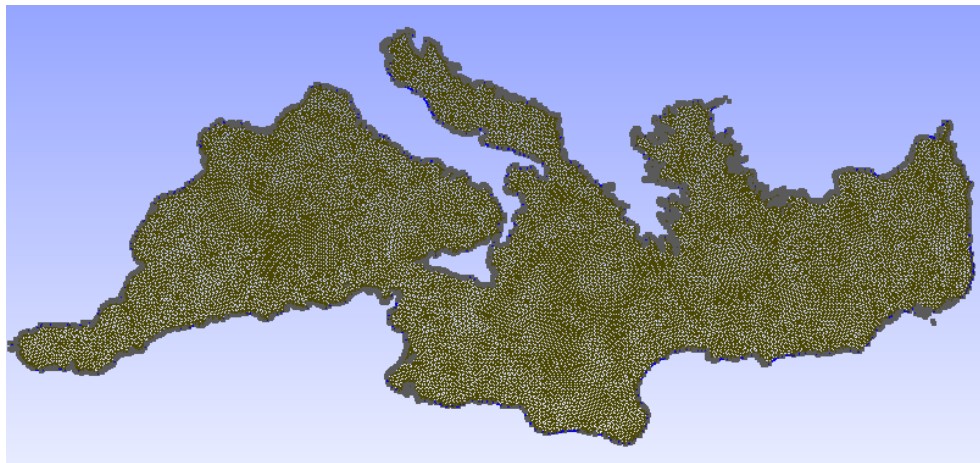

**Figure 2.** Triangulation of the Mediterranean Sea suitable for simulations with finite element methods which was used in [30,60] (see also [61–63]).

In [34,67], Medjo considered this formulation and proved bounds for the time discretization error. Cascon et al. [68] proved both a priori and a posteriori error estimates for the FE discretization of the linear Stommel-Munk model, which is a simplified version of the QGE obtained by dropping the nonlinear term.

The streamfunction formulation (13) of QGE is a fourth-order PDE, which naturally necessitates $C^1$ elements for a conforming finite element discretization. Considering the finite element space $W \subset H_0^2(\Omega)$, the finite element discretization reads: Find $\psi_h \in W$ [66] satisfying

$$
\left(\frac{\partial}{\partial t} \nabla \psi_h, \nabla \phi_h\right) + \mathrm{Re}^{-1}(\Delta \psi_h, \Delta \phi_h) + (J(\psi_h, \Delta \psi_h), \phi_h) - \mathrm{Ro}^{-1}\left(\frac{\partial \psi_h}{\partial x}, \phi_h\right) = \mathrm{Ro}^{-1}(F_e, \phi_h), \forall \phi_h \in W.
\tag{17}
$$

To our knowledge, the first optimal error convergence results for the finite element approximation of the QGE (12) were proved for the streamfunction formulation (17) using Argyris elements in [30]. Several numerical tests, commonly employed in the geophysical literature, showed the accuracy of the finite element discretization and illustrated the theoretical estimates. Other recent developments of FEM for QGE include discontinuous Galerkin formulation using $C^0$ elements [62] and B-splines [69–73]. In particular, an adaptive refinement algorithm for B-splines finite element approximation was presented in [71] for the streamfunction formulation.

## 4. Reduced Order Models (ROMs)

ROMs for the QGE have been developed for decades (see, e.g., [8,32,61,74–83]). Most ROMs have been constructed by using a classical Galerkin projection framework, but data-driven modeling (e.g., machine learning) has also been recently used [84,85]. In Section 4.1, we outline the standard Galerkin ROM construction. In Section 4.2, we explain the importance of considering under-resolved regimes when developing ROMs for realistic, chaotic flows. Furthermore, we present several ROM closure strategies, which are generally needed when ROMs are used in an under-resolved regime. The flowchart of the ROMs presented in this section is illustrated in Figure 3.

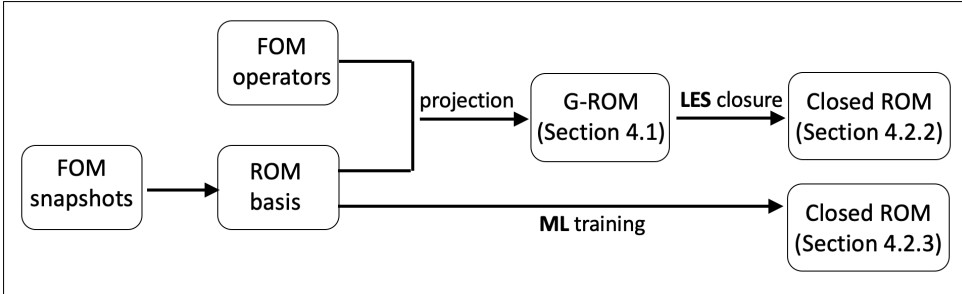

**Figure 3.** Framework of the ROMs presented in Section 4.

*4.1. Galerkin Reduced Order Model (G-ROM)*

To construct the standard Galerkin ROM, we start by generating the ROM basis. To this end, we use the proper orthogonal decomposition (POD) [2,6], which is also known as empirical orthogonal functions (EOF) and principal component analysis (PCA). We emphasize, however, that other ROM bases could be used, such as principal interaction patterns (PIPs) and optimal persistence patterns (OPPs) [74] (see also [1,3,5,7,75,86,87] for alternative strategies).

The POD starts by collecting the snapshots $\{\omega_h^1, \ldots, \omega_h^M\}$, which are numerical approximations of the vorticity in the QGE (15) at $M$ different time instances. We consider relatively accurate snapshots. If inaccurate snapshots are used to construct the POD basis, the resulting ROM can be inaccurate (see, e.g., [88]). For clarity of presentation, in this paper we use the finite element discretization, but other numerical discretizations could be used. The POD seeks a low-dimensional basis that approximates the snapshots optimally with respect to a certain norm. In this presentation, we use the $L^2$ norm and the $L^2$ inner product:

$$\left(\omega_1, \omega_2\right) = \int_\Omega \omega_1(\mathbf{x})\, \omega_2(\mathbf{x})\, d\mathbf{x}. \tag{18}$$

We note that, although the $L^2$ norm and the $L^2$ inner product are the most popular choices in reduced order modeling, other norms and inner products could also be used (see, e.g., [89]). To construct POD basis functions that approximate the snapshots optimally with respect to the $L^2$ norm, we solve the following minimization problem [89]:

$$\min_{\widetilde{\varphi}_1, \cdots, \widetilde{\varphi}_N} \sum_{j=1}^{M} \left\| \omega_h^j - \sum_{i=1}^{N} \left(\omega_h^j, \widetilde{\varphi}_i\right) \widetilde{\varphi}_i \right\|_{L^2}^2 \tag{19}$$
$$\text{s.t. } \left(\widetilde{\varphi}_l, \widetilde{\varphi}_m\right) = \delta_{lm} \quad \text{for } 1 \leq l, m, \leq N,$$

where $\delta_{lm}$ is the Kronecker delta. The solution of the minimization problem (19) is equivalent to the solution of the eigenvalue problem

$$Y^T M_h Y \widetilde{\varphi}_j = \lambda_j \widetilde{\varphi}_j, \quad j = 1, \ldots, N, \tag{20}$$

where $Y$ denotes the snapshot matrix, whose columns correspond to the finite element coefficients of the snapshots, $M_h$ denotes the finite element mass matrix, and $N$ is the dimension of the finite element space. The eigenvalues are real and non-negative, so they can be ordered as follows: $\lambda_1 \geq \lambda_2 \geq \ldots \geq \lambda_R \geq \lambda_{R+1} = \ldots = \lambda_N = 0$, where $R$ is the rank of the snapshot matrix. It can be shown [89] that these eigenvalues determine how well the corresponding POD modes represent the given vorticity snapshots: the lower the eigenvalue index, the more important the corresponding POD mode. Thus, we choose the POD vorticity basis functions $\{\varphi_j\}_{j=1}^r$ from the eigenfunctions in (20) that correspond to the first $r \leq R$ largest eigenvalues and define the ROM vorticity space as $X^r := \text{span}\{\varphi_1, \ldots, \varphi_r\}$.

To determine the POD streamfunction basis functions, we use the POD vorticity basis functions and follow the approach in [8,32]. Specifically, we define the POD streamfunction basis functions as the normalized functions $\{\phi_j\}_{j=1}^r$, which are chosen such that the satisfy the following Poisson problem with homogeneous Dirichlet boundary conditions:

$$-\Delta\phi_j = \varphi_j, \quad j = 1,\ldots,r. \tag{21}$$

Next, we define the ROM approximations of the vorticity and streamfunction as follows:

$$\omega_r(\boldsymbol{x},t) = \sum_{j=1}^r a_j(t)\,\varphi_j(\boldsymbol{x}), \tag{22}$$

$$\psi_r(\boldsymbol{x},t) = \sum_{j=1}^r a_j(t)\,\phi_j(\boldsymbol{x}), \tag{23}$$

where $\{a_j(t)\}_{j=1}^r$ are the sought time-varying ROM coefficients. We note that we made two important choices in our approach: (i) We enforced the coupling between the POD vorticity and streamfunction basis functions in (21); and (ii) We used the same ROM coefficients in the ROM vorticity approximation (22) and in the ROM streamfunction approximation (23). The motivation for making these two choices is efficiency. Indeed, we only need to construct a ROM for the vorticity; once the coefficients $a_j$ are determined from (15a), Equation (15b) is automatically satisfied. (Of course, one could use a different approach and construct two different ROM bases and two different ROM approximations for the vorticity and streamfunction, but that would increase the ROM computational cost.) To construct a ROM for the vorticity, we replace the vorticity $\omega$ by $\omega_r$ in the QGE (15a), and then we use a Galerkin projection onto $X^r$. Thus, we obtain the Galerkin ROM (G-ROM) for the QGE: $\forall\, i = 1,\ldots,r$,

$$\left(\frac{\partial\omega_r}{\partial t},\varphi_i\right) + (J(\omega_r,\psi_r),\varphi_i) - \text{Ro}^{-1}\left(\frac{\partial\psi_r}{\partial x},\varphi_i\right) + \text{Re}^{-1}(\nabla\omega_r,\nabla\varphi_i) = \text{Ro}^{-1}\left(F_e,\varphi_i\right). \tag{24}$$

The G-ROM (24) yields the following autonomous dynamical system for the vector of time coefficients, $\mathbf{a}(t) = (a_i(t))_{i=1,\ldots,r}$:

$$\overset{\bullet}{\mathbf{a}} = \mathbf{b} + \mathbf{A}\,\mathbf{a} + \mathbf{a}^\top\mathbf{B}\,\mathbf{a}, \tag{25}$$

where $\mathbf{b}$, $\mathbf{A}$, and $\mathbf{B}$ are an $r \times 1$ vector, an $r \times r$ matrix, and an $r \times r \times r$ tensor, which correspond to the constant, linear, and quadratic terms in the numerical discretization of the QGE (15), respectively. The $r$-dimensional system (25) can be written componentwise as follows: For all $i = 1,\ldots,r$,

$$\overset{\bullet}{a}_i(t) = b_i + \sum_{m=1}^r A_{im}a_m(t) + \sum_{m=1}^r\sum_{n=1}^r B_{imn}\,a_m(t)\,a_n(t), \tag{26}$$

where

$$b_i = \text{Ro}^{-1}\left(F_e,\varphi_i\right), \tag{27}$$

$$A_{im} = \text{Ro}^{-1}\left(\frac{\partial\phi_m}{\partial x},\varphi_i\right) - Re^{-1}\left(\nabla\varphi_m,\nabla\varphi_i\right), \tag{28}$$

$$B_{imn} = -\left(J(\varphi_m,\phi_n),\varphi_i\right). \tag{29}$$

The G-ROM (25) has been investigated in the numerical simulation of the QGE (15) (see, e.g., [8,32,80,83]), where it was shown that it can decrease the FOM computational cost

by orders of magnitude. However, the numerical simulations in [8,32] have also shown that a low-dimensional G-ROM is not able to produce accurate approximations of the streamfunction and the velocity fields. The G-ROM's numerical inaccuracy in [8,32] is due to the lack of a closure model [8,9,90], which we discuss in Section 4.2.

### 4.2. ROM Closure Models

In this section, we survey the *ROM closure models* developed for the QGE (12). First, we define closure modeling and we explain why it is needed when ROMs are used in the under-resolved regime (Section 4.2.1). Then, we present the two main types of ROM closure modeling for the QGE that are in current use: large eddy simulation (LES) ROM closure models (Section 4.2.2) and machine learning (ML) ROM closure models (Section 4.2.3). While LES and ML ROM closures are both data-driven modeling approaches, they are different in the way they use data to develop a closure model: The LES approach is based on ROM spatial filtering and least squares methods, whereas the ML approach is based on machine learning techniques.

#### 4.2.1. Under-Resolved ROMs Require Closure Models

The concept of under-resolved simulations is central in classical CFD. *Under-resolved simulations are those simulations in which the number of degrees of freedom* (e.g., the number of mesh points or basis functions) *is not enough to capture the dynamics of the underlying system.* For example, in turbulent flow simulations the available number of mesh points in a finite element or finite volume discretization, or the number of basis functions in a spectral discretization are not enough to resolve all the lengthscales in the turbulent flow, down to the Kolmogorov scale [91–93]. The numerical simulations at these inherently coarse resolutions are called under-resolved simulations.

In under-resolved simulations of turbulent flows, standard discretizations yield inaccurate results, which are not acceptable in practical engineering settings, e.g., large relative errors, inaccurate quantities of interest (e.g., lift and drag), and inaccurate flow features (e.g., vortex shedding frequency for the flow past a cylinder). In these cases, the classical computational models (e.g., the Navier–Stokes equations) are generally supplemented with correction terms that model the effect of the neglected scales (e.g., the scales smaller than the given coarse mesh size). These correction terms are generally called *closure models* [91–93].

The concept of under-resolved simulations is also relevant to reduced order modeling: *Under-resolved ROM simulations are those simulations in which the ROM dimension is not enough to capture the dynamics of the underlying system.* But how exactly do we determine whether a ROM simulation is resolved or under-resolved? Next, we present several potential answers to this question. Some of these answers are *a priori* criteria (i.e., can be used before the ROM simulation), some are *a posteriori* criteria (i.e., can be used only after the ROM simulation).

*Kolmogorov n-width*: The Kolmogorov n-width is an *a priori* criterion to determine whether the ROM simulation is resolved or under-resolved. Given the solution manifold $\mathcal{M}$ of the underlying system's dynamics, the Kolmogorov n-width [94] provides a way to quantify the best *n*-dimensional trial subspace $\mathcal{X}^n$:

$$d_n(\mathcal{M}) := \inf_{\mathcal{X}^n} \sup_{\omega \in \mathcal{M}} \inf_{g \in \mathcal{X}^n} \|\omega - g\|.$$

Of course, calculating the Kolmogorov n-width for general systems can be challenging. There are, however, cases when the relative size of the Kolmogorov n-width is known. For example, it is known that, for computational problems dominated by diffusion, the Kolmogorov n-width decays fast, while for those dominated by convection, it decays slowly [95]. As a result, in order to obtain an accurate approximation of the solution manifold, the dimension of the ROM trial space is expected to be much higher in the convection-dominated case than in the diffusion-dominated case. Thus, for convection-dominated systems, if we use a very high-dimensional (i.e., of the same order as the Kolmogorov n-width) ROM, we obtain a resolved ROM simulation. If, however, we use

a low-dimensional (i.e., much lower than the Kolmogorov n-width) ROM, we obtain an under-resolved ROM simulation.

*Eigenvalue decay rate*: The eigenvalue decay rate is an *a priori* criterion to determine whether the ROM simulation is resolved or under-resolved. The eigenvalues $\lambda_1, \ldots, \lambda_R$ in the eigenvalue problem (20) (used to construct the ROM basis) represent the energy content of the corresponding ROM modes [2,89]. Thus, the ratio

$$\frac{\sum_{i=1}^{r} \lambda_i}{\sum_{i=1}^{R} \lambda_i} \tag{30}$$

defines the relative energy content of the first *r* ROM basis functions with respect to the total energy of the system (see, e.g., page 16 in [89]). We emphasize that the concept of "energy" in this context is used in a generic sense. For example, when the snapshots are FOM approximations of a the velocity field in a fluid flow, the energy in (30) is the kinetic energy; when the snapshots are FOM approximations of the vorticity field in the QGE, the energy in (30) is the enstrophy. We can define the resolved regime as the regime in which the ROM dimension *r* is large enough to ensure that the relative energy ratio (30) is larger than a certain threshold (e.g., 90%). Thus, we expect a low-dimensional ROM to be in the resolved regime when the eigenvalues have a fast decay, and in the under-resolved regime when the eigenvalues have a slow decay.

To illustrate this point, in Figure 4 we plot the scaled eigenvalues $\lambda_k/\lambda_1$, $k = 1, \ldots, 150$ for two flow settings: the 2D flow past a cylinder at Re = 1000 and the QGE with Re = 450 and Ro = 0.0036 (the latter will be used in the numerical investigation in Section 5). This plot shows that the eigenvalues decay much faster for the flow past a cylinder case than for the QGE case.

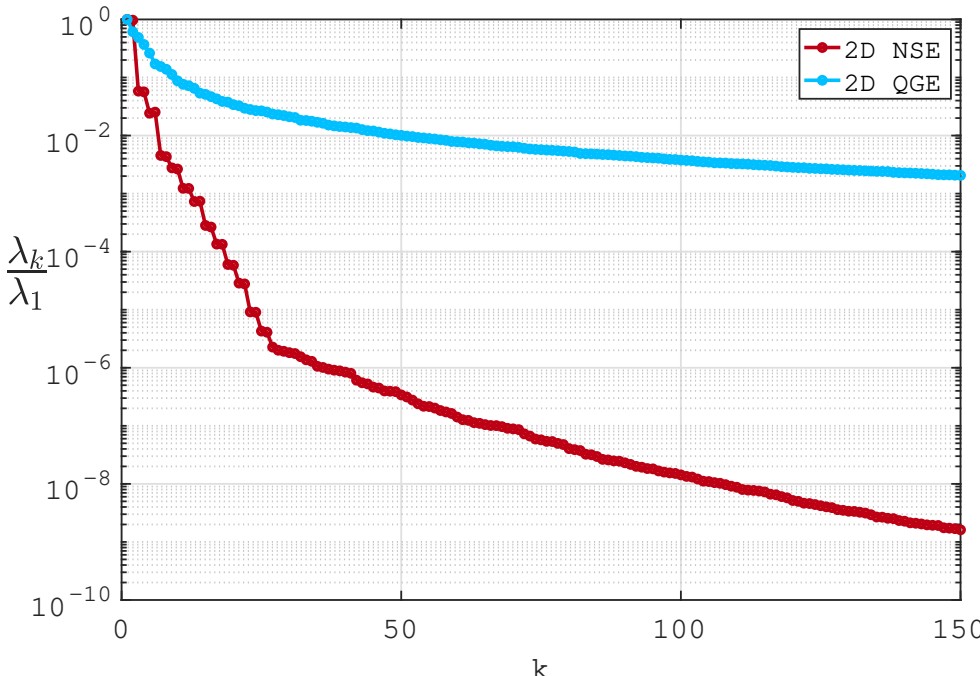

**Figure 4.** Scaled eigenvalues $\frac{\lambda_k}{\lambda_1}$ for the 2D flow past a circular cylinder with Re = 1000 and the QGE with Re = 450 and Ro = 0.0036 (see Section 5 for details).

Indeed, the results in Table 1 show that, in order to achieve a 90% relative energy ratio in (30), we need to use only 2 ROM modes for the flow past a cylinder, and 77 ROM modes for the QGE case. Thus, if we use only a handful of ROM modes to ensure a low computational cost, we expect the resulting low-dimensional ROM to accurately capture the dynamics of the flow past a cylinder, but not the dynamics of the QGE. In this case, we

perform a resolved ROM simulation of the flow past a cylinder, and an under-resolved ROM simulation for the QGE.

**Table 1.** Number of ROM modes needed to achieve a given relative energy content (30) for the 2D flow past a circular cylinder with Re = 1000 and the QGE with Re = 450 and Ro = 0.0036 (see Section 5 for details).

| Relative Energy Content | 90% | 95% | 99% |
|---|---|---|---|
| 2D flow past a cylinder | 2 | 4 | 6 |
| QGE | 77 | 152 | 380 |

*ROM Lengthscale*: The ROM lengthscale is an *a priori* criterion to determine whether the ROM simulation is resolved or under-resolved. In principle, the ROM lengthscale criterion follows the same algorithm as the standard CFD lengthscale criterion: Start with a lengthscale that is large enough to capture the relevant dynamics, and then choose the input discretization parameters such that phenomena occurring at the chosen lengthscale can be approximated. Choosing the discretization parameters is, however, fundamentally different in classical CFD and ROMs: In classical CFD, the *spatial meshsize* (e.g., for finite difference or finite element methods) or the cutoff wavenumber in a Fourier truncation (e.g., for spectral methods) clearly determines what lengthscale can be approximated. For ROMs, however, *there is no straightforward definition of a lengthscale based on the ROM discretization parameters*, i.e., the ROM dimension ($r$), the ROM basis ($\{\varphi_1, \ldots, \varphi_r\}$), and the ROM eigenvalues ($\{\lambda_1, \ldots, \lambda_r\}$). To our knowledge, only very few ROM lengthscale definitions based on the ROM discretization parameters have been proposed. In [96], a ROM lengthscale was defined for the 3D flow past a circular cylinder at $Re = 1000$ (see also [2] for related work). This lengthscale was then used in [96] to build ROM closure models.

*Trial and error*: The trial and error approach is an *a posteriori* criterion to determine whether the ROM simulation is resolved or under-resolved. Specifically, a few ROM simulations are run in the offline stage in order to determine the ROM discretization parameters that yield accurate results, which are acceptable in practical engineering settings, e.g., small relative errors, accurate quantities of interest (e.g., lift and drag), and accurate flow features (e.g., vortex shedding frequency for the flow past a cylinder).

In Section 5, we show that under-resolved ROM simulations of the QGE can yield inaccurate results. To increase the accuracy of these under-resolved ROM simulations, the standard G-ROM (25) is generally supplemented with a closure model:

$$\dot{\mathbf{a}} = \mathbf{b} + \mathbf{A}\,\mathbf{a} + \mathbf{a}^\top \mathbf{B}\,\mathbf{a} + \boldsymbol{\tau}^{ROM}, \tag{31}$$

where $\boldsymbol{\tau}^{ROM}$ is the closure model that needs to be determined.

There are two main types of ROM closure modeling approaches, i.e., approaches to modeling the term $\boldsymbol{\tau}^{ROM}$ in (31) in the offline stage:

- **Black box** ROM closure models: These models consider the true closure model $\boldsymbol{\tau}^{FOM}$ as a black box, i.e., the specific form of $\boldsymbol{\tau}^{FOM}$ is not determined. Instead, one first postulates a model form for $\boldsymbol{\tau}^{FOM}$, i.e., $\boldsymbol{\tau}^{FOM} \approx \boldsymbol{\tau}^{ROM}$, and then determines the parameters of the model form $\boldsymbol{\tau}^{ROM}$, either by using available data or physical insight.

- **Mathematical** ROM closure models: These models use *filtering/averaging* (e.g., with respect to space, time, or initial conditions) to determine the specific form of the true ROM closure term $\boldsymbol{\tau}^{FOM}$. As in the black box ROM closure models, one postulates a model form for $\boldsymbol{\tau}^{FOM}$, i.e., $\boldsymbol{\tau}^{FOM} \approx \boldsymbol{\tau}^{ROM}$. However, the mathematical ROM closure modeling utilizes *data for the specific form* of $\boldsymbol{\tau}^{FOM}$ to determine the ROM closure model $\boldsymbol{\tau}^{ROM}$.

In this paper, we do not survey the ROM closure models. Instead, we only focus on ROM closure modeling for the QGE. Specifically, in Sections 4.2.2 and 4.2.3, we present two

different ROM closure modeling strategies for the QGE. We note, however, that there are alternative ROM closure modeling strategies for the QGE, e.g., the stochastic mode reduction strategy developed by Majda and his collaborators (see [76] and references therein).

### 4.2.2. Large Eddy Simulation ROM Closure Models

The large eddy simulation (LES) ROM closure modeling is inspired from classical LES of turbulent flows [91–93]. The LES-ROM closure models come in two flavors: black box and mathematical.

The black box LES-ROM closure models developed for the QGE use physical insight to postulate a model form for the closure term. Specifically, they postulate that the ROM closure term has to be dissipative. In [80], a linear damping term (i.e., a third-order Laplace operator) is used as a ROM closure model. In [32], a nonlinear damping term (i.e., a simplified Smagorinsky model [97]) is used as a ROM closure model. A significant improvement to the Smagorinsky ROM closure model used in [32] is the dynamic subgrid-scale ROM closure model which was first proposed in [96] and later adapted to the QGE in [79].

The mathematical LES-ROM closure models developed for the QGE are an elegant approach to closure modeling. These ROM closure models are built in three steps: In the first step, the QGE are *spatially filtered* to obtain the large structures which can be approximated at the given coarse resolution. In this first step, an exact formula for the ROM closure term $\boldsymbol{\tau}^{FOM}$ is also obtained. In the second step, a specific model form is postulated for the LES-ROM closure model, i.e., $\boldsymbol{\tau}^{FOM} \approx \boldsymbol{\tau}^{ROM}$ in (31). Finally, in the third step, FOM data is used to find the parameters in the general form $\boldsymbol{\tau}^{ROM}$ that yield the closest (in a least squares sense) approximation to true ROM closure term $\boldsymbol{\tau}^{FOM}$. One example of mathematical LES-ROMs is the recently developed *data-driven variational multiscale* ROM (DD-VMS-ROM) [8,9,90], which is centered around the variational multiscale framework. There are two versions of the DD-VMS-ROM: a two-scale model [8,9] and an improved three-scale model [90]. For clarity of presentation, we present the two-scale DD-VMS-ROM. To construct the DD-VMS-ROM, the ROM projection from the ROM space $X^R = \text{span}\{\varphi_1, \ldots, \varphi_R\}$ to the subspace $X^r = \text{span}\{\varphi_1, \ldots, \varphi_r\}$, $r \leq R$ is used as a spatial filter. The filtered QGE yield an exact formula for the ROM closure term, $\boldsymbol{\tau}^{FOM}$. Next, a specific model form is prescribed for the exact closure term

$$\boldsymbol{\tau}^{FOM} \approx \boldsymbol{\tau}^{ROM} := \widetilde{A}\,\boldsymbol{a} + \boldsymbol{a}^\top \widetilde{B}\,\boldsymbol{a} \tag{32}$$

and the entries of the LES-ROM closure operators $\widetilde{A}$ and $\widetilde{B}$ are found by solving the following *least squares problem*:

$$\min_{\widetilde{A}, \widetilde{B}} \sum_{j=1}^{M} \left\| \boldsymbol{\tau}^{FOM}(t_j) - \left( \widetilde{A}\,\boldsymbol{a}^{FOM}(t_j) + (\boldsymbol{a}^{FOM}(t_j))^\top \widetilde{B}\,\boldsymbol{a}^{FOM}(t_j) \right) \right\|^2, \tag{33}$$

where $\boldsymbol{a}^{FOM}$ are computed from the FOM data. Finally, the LES-ROM closure operators obtained in (33) are used to build the DD-VMS-ROM:

$$\overset{\bullet}{\mathbf{a}} = \mathbf{b} + \left( \mathbf{A} + \widetilde{A} \right) \mathbf{a} + \mathbf{a}^\top \left( \mathbf{B} + \widetilde{B} \right) \mathbf{a}. \tag{34}$$

In Section 5, we investigate the DD-VMS-ROM in the under-resolved simulation of the QGE.

### 4.2.3. Machine Learning ROM Closure Models

Machine learning (ML) methods have recently started to make an impact in reduced order modeling of fluid flows. The ML methods most frequently used to build ROMs include multilayer perceptron (MLP) [85,98,99], convolution neural networks (CNN) [100], recurrent neural networks (RNN) [84,101,102], and variational autoencoder (VAE) [100,103]. Some of these ML-ROMs are *nonintrusive*, i.e., they use the FOM codes as black boxes,

only to generate output data from different inputs. For these nonintrusive ML-ROMs, no prior information about the underlying governing equations is required to construct the model. These models fully rely on data combined with ML methods to discover the ROM dynamics and can be written as follows:

$$\dot{\boldsymbol{a}} = \widehat{\boldsymbol{F}}(\boldsymbol{a}, \boldsymbol{\theta}), \tag{35}$$

where $\boldsymbol{a}$ is the vector of ROM coefficients and $\boldsymbol{\theta}$ is the vector of learnable parameters in the ML model $\widehat{\boldsymbol{F}}$. The nonintrusive ML-ROMs are fundamentally different from classical intrusive modeling strategies, such as the Galerkin method used to generate the G-ROM (25), which need access to the underlying governing equations in order to construct the ROM.

Only few ML-ROMs have been developed for the QGE. For example, an extreme learning machine concept with neural networks was introduced for the ROM closure of the QGE in [85]. Furthermore, a nonintrusive reduced order modeling framework embedded with a long short-term memory (LSTM) network was developed for quasi-geostrophic turbulence to improve the time series prediction of ROMs in [84]. The LSTM-ROM for QGE [84] was constructed (trained) in two steps:

1. The ROM coefficients in a given time window $\{\boldsymbol{a}^{FOM,(n-k)}, \boldsymbol{a}^{FOM,(n-k+1)}, \dots, \boldsymbol{a}^{FOM,(n)}\}$ were extracted from the high-resolution FOM data by projecting the snapshots onto the ROM modes.

2. The LSTM neural network was used to construct an ML-ROM that mapped the old ROM coefficients $\{\boldsymbol{a}^{FOM,(n-k)}, \boldsymbol{a}^{FOM,(n-k+1)}, \dots, \boldsymbol{a}^{FOM,(n)}\}$ to the ROM coefficients at the new time step $\boldsymbol{a}^{FOM,(n+1)}$.

The resulting model was then used in the testing stage to predict the ROM coefficients at new time instances.

Recently, *hybrid* ROMs that combine classical Galerkin modeling with machine learning have started to become popular. For example, a hybrid ROM closure was proposed in [104] for the QGE. This hybrid ROM combined classical Galerkin projection methods with neural network closures to perform near real-time prediction of mesoscale ocean flows. The numerical investigation in [104] showed that the hybrid ROM was more accurate than both the classical G-ROM and a pure ML-ROM (i.e., a ROM built entirely from data by using machine learning).

## 5. Numerical Results

In this section, we present an illustration of the projection ROMs constructed in Section 4 in the numerical simulation of the QGE described in Section 2. First, we describe the details of the computational setting that we use in our ROM numerical investigation: the regimes (Section 5.1), the test problem (Section 5.2), the criteria (Section 5.3), and the generation of the FOM data used to construct the ROM basis (Section 5.4). After we clarify these details, we perform a numerical investigation of the ROM accuracy and ROM efficiency (Section 5.5).

### 5.1. Regimes

In our numerical illustration, we use four regimes: (i) a *reconstructive* regime, which is an easier test case, in which the ROM is validated on the same time interval as the time interval used to train the ROM; (ii) a *predictive* regime, which is a harder test case, in which the ROM is trained on a short time interval and validated on a longer time interval; (iii) a *resolved* regime, in which the number of ROM basis functions is enough to represent the system's dynamics; and (iv) an *under-resolved* regime, in which the number of ROM basis functions is not enough to represent the system's dynamics. These four regimes illustrate different features of the ROMs.

The reconstructive regime is the first step in a ROM investigation. At the very least, the proposed ROM needs to provide an efficient and accurate approximation of the FOM data used to train it (i.e., the FOM results used to construct the ROM basis). The predictive

regime is a harder test in the ROM investigation. In order to be practical, the proposed ROM needs to be able to approximate the FOM results on time intervals and parameter ranges that are wider than those used to train the ROM. (For clarity, in this section, we consider only a longer time interval, but wider parameter ranges could also be considered.) Of course, the proposed ROM generally has a harder time approximating data that it has not seen in the training process, but the ROM needs to perform well in the predictive regime in order to be deemed successful in practice.

The resolved regime is an easier test in the ROM investigation. Since the ROM uses a relatively large number of ROM basis functions, which is enough to capture the underlying system's dynamics, a straightforward, standard G-ROM is expected to perform well in the resolved regime. The under-resolved regime is a much harder test in the ROM investigation. In the under-resolved regime, the proposed ROM needs to use a relatively small (i.e., not enough to capture the system's dynamics) number of ROM basis functions and somehow still be able to approximate the FOM data. In classical computational fluid dynamics (CFD), the under-resolved regime is one of the most important tests for the practicality of the proposed numerical method. Indeed, many realistic CFD applications are turbulent and chaotic, and standard resolved discretizations (e.g., direct numerical simulation (DNS)) are simply not possible, since they require an unrealistic number of degrees of freedom. The under-resolved regime is relatively much less investigated in the ROM world. We believe, however, that to develop ROMs that can be used in the numerical simulation of *realistic, chaotic* geophysical flows, the proposed ROMs need to be investigated in the under-resolved regime.

Since the reconstructive and predictive regimes, on the one hand, and the resolved and under-resolved regimes, on the other hand, serve different purposes, we consider four regime pairs in the ROM numerical investigation in Section 5.5: First, we consider the resolved and reconstructive, and the resolved and predictive regimes. The goal here is to investigate the reconstructive and, more importantly, the predictive capabilities of the standard G-ROM in the relatively simple resolved regime. Second, we consider the under-resolved and reconstructive, and the under-resolved and predictive regimes. The goal here is different. We want to investigate the reconstructive and predictive capabilities of the standard G-ROM in the challenging under-resolved regime. We expect that, when only a few ROM basis functions are used to build it, the standard G-ROM will perform poorly in the under-resolved regime. Thus, to address the G-ROM's potential inaccuracies, we also consider the LES-ROM proposed in Section 4.2.2, i.e., the DD-VMS-ROM. In the under-resolved regime, we expect the LES-ROM to be more accurate than the standard G-ROM.

### 5.2. Test Problem Setup

In our ROM numerical investigation in a QGE setting, we need to make several choices. Specifically, in the QGE (15), we need to choose the spatial domain, the time interval, the forcing ($F_e$), the Reynolds number (Re), and the Rossby number (Ro). We emphasize that these choices are important: Some choices yield a relatively easy test problem, i.e., a problem in which a standard ROM built with relatively few ROM basis functions can generate an accurate and efficient approximation. Other choices, however, yield a challenging test problem, in which standard low-dimensional ROMs produce inaccurate results.

In our numerical investigation, we choose parameters that yield a *challenging test problem*, which has been used in numerous studies (see, e.g., [8,27–29,31–33,79,85,105]) as a simplified model for more realistic ocean dynamics. Specifically, we choose the simple spatial domain $[0,1] \times [0,2]$, the relatively long time interval $[0,100]$, and a symmetric double-gyre wind forcing given by $F_e = \sin(\pi(y-1))$, which yields a four-gyre circulation in the time mean. We also choose the same Reynolds number and Rossby number as those used in [8,28,32,33], i.e., Re = 450 and Ro = 0.0036.

We emphasize that this four-gyre QGE test problem represents a significant challenge for FOM simulations with standard numerical methods. Indeed, as shown in [27], although a double-gyre wind forcing is used, the long term time-average yields a *four-gyre* pattern (see Figure 5). On realistic coarse meshes, classical numerical methods (e.g., finite element and finite volume methods) generally produce inaccurate approximations to this test problem. In particular, standard numerical discretizations fail to recover the correct four-gyre pattern (see, e.g., [32,33]). One of the main reasons for the challenging character of the four-gyre test problem is the relatively low Rossby number used (i.e., $Ro = 0.0036$). Indeed, as shown in Figure 1, a relatively small Rossby number yields a sharp western boundary layer, which makes the test problem challenging for FOM simulations (see, e.g., [32,33]). While the Reynolds number used (i.e., $Re = 450$) is not large by turbulence modeling standards, it turns out that it yields a convection-dominated regime that is challenging for FOM simulations. Overall, these parameter choices together with the chosen spatial domain, time interval, and forcing function, yield a challenging FOM test problem. This is clearly illustrated in the plot of the FOM kinetic energy in Figure 6, which suggests that this is a *chaotic* system, with non-periodic time evolution.

Given the non-periodic, chaotic evolution of the this four-gyre test problem, we expect it to represent a challenging test not only for FOM simulations, but also for ROM simulations. This expectation is supported by projecting the FOM data on the ROM basis functions to obtain the true ROM coefficients, which the proposed ROMs need to approximate. These true ROM coefficients, which are plotted in Figure 7, have a non-periodic, chaotic evolution, which is challenging to capture by standard ROMs. In Section 5.5, we will show that this four-gyre test problem does indeed represent a challenging test for ROMs.

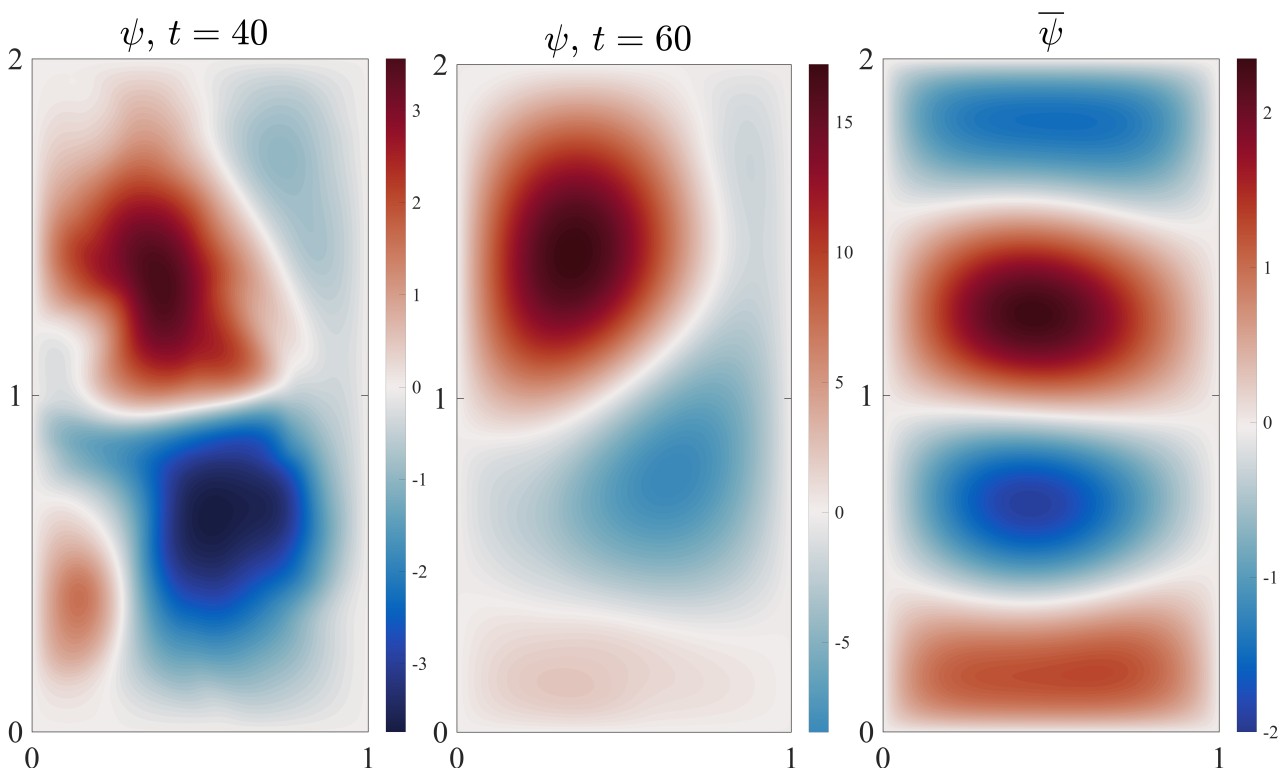

**Figure 5.** FOM streamfunction contour plots at $t = 40$ (**left**), $t = 60$ (**middle**), and time-averaged (**right**).

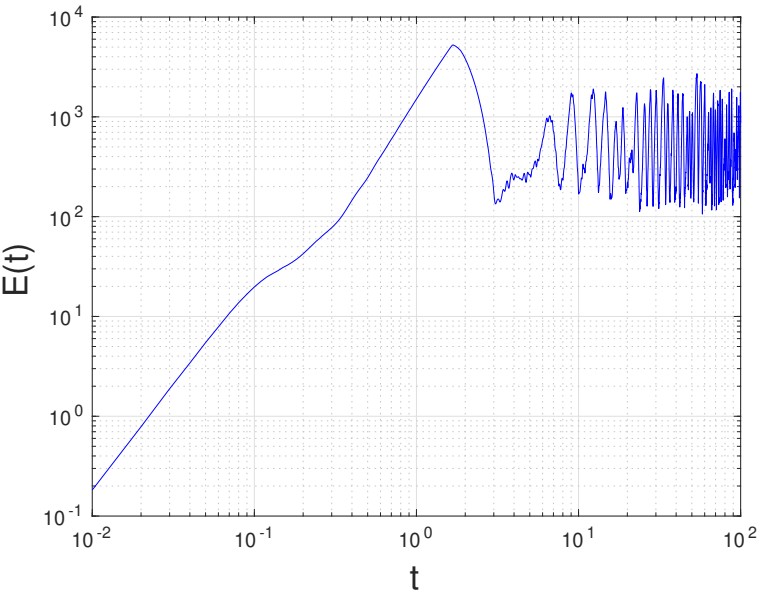

**Figure 6.** Time evolution of the kinetic energy of the FOM.

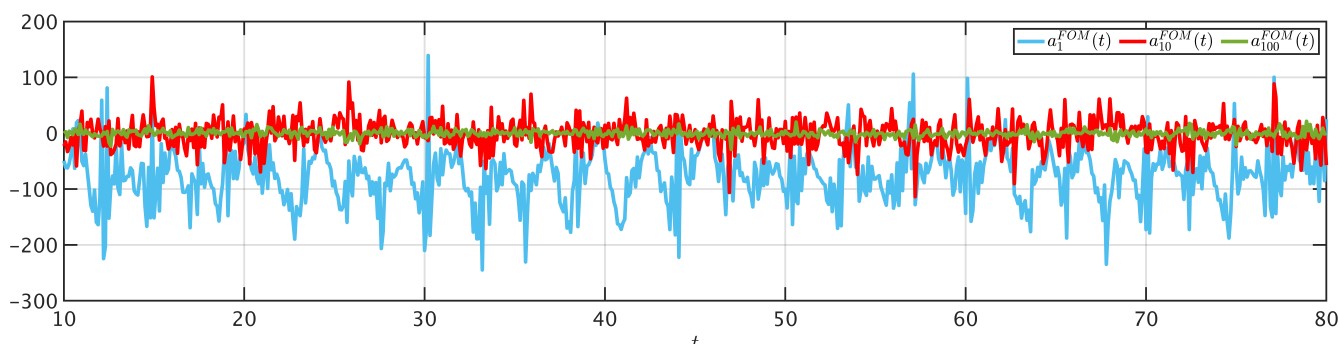

**Figure 7.** Time evolution of $a_1^{FOM}(t), a_{10}^{FOM}(t), a_{100}^{FOM}(t)$.

**Remark** (QGE vs. 2D Flow Past a Cylinder). *The 2D flow past a cylinder at low Reynolds numbers has become one of the most popular test problems in the ROM world. The reason is that the time evolution of the true ROM coefficients is periodic and a few ROM modes are required to capture the system's dynamics. By comparison, the QGE test problem used in our numerical investigation is a significantly harder test problem: Its true ROM coefficients display a non-periodic, chaotic time evolution and relatively many ROM modes are required to capture the system's dynamics. This statement is supported by the plot in Figure 4 and the results in Table 1: The plot shows that the eigenvalues decay much faster for the flow past a cylinder test case than for the QGE test case. The results in Table 1 show that, in order to achieve a 90% relative energy content (which is defined in (30)), the flow past a cylinder test case requires only 2 ROM modes, whereas the QGE test case requires 77 ROM modes.*

### 5.3. Criteria

To investigate the ROMs, we use the following three criteria: (i) the relative $L^2$ norm of the time-averaged streamfunction errors between $\psi^{FOM}$ and $\psi^{ROM}$:

$$\left\| \frac{1}{M} \sum_{j=1}^{M} \psi^{FOM}(t_j) - \frac{1}{M} \sum_{j=1}^{M} \psi^{ROM}(t_j) \right\|_{L^2}^2 \bigg/ \left\| \frac{1}{M} \sum_{j=1}^{M} \psi^{FOM}(t_j) \right\|_{L^2}^2 . \tag{36}$$

(ii) The ROM's ability to recover the four-gyre pattern of the time-average of the FOM streamfunction in Figure 5. (iii) The ROM computational cost. The first two criteria quantify the ROM numerical accuracy, whereas the third criterion quantifies the ROM efficiency. We note that the first two criteria utilize time-averages. The reason for using time-averages is that, in the numerical investigation of chaotic systems (such as the four-gyre test problem), pointwise in time quantities are less robust (e.g., prone to phase errors) and can yield deceiving results.

   To define the resolved and under-resolved ROM regimes, we use two of the four criteria outlined in Section 4.2.1: (i) the trial and error criterion; and (ii) the eigenvalue decay rate criterion. Specifically, when we use the trial and error criterion in our numerical investigation, we call the ROM regime resolved if its relative $L^2$ norm of the error (36) is $\mathcal{O}(10^{-1})$, and under-resolved otherwise. We note that an $\mathcal{O}(10^{-1})$ relative error is large by engineering standards. However, our numerical investigation will show that even this large threshold requires high-dimensional ROMs. When we use the eigenvalue decay rate criterion in our numerical investigation, we call the ROM regime resolved if its relative kinetic energy content (which was defined in (30)) is above 90%, and under-resolved otherwise.

*5.4. FOM Snapshot Generation*

   To generate the FOM data (i.e., snapshots) that is used to construct the ROM basis functions, we utilize fine resolution spatial and temporal discretizations. Specifically, for the FOM spatial discretization, we use a pseudospectral method with a $257 \times 513$ spatial resolution [8]. For the FOM time discretization, we use an explicit Runge-Kutta method (Tanaka-Yamashita, an order 7 method with an embedded order 6 method for error control), and an error tolerance of $10^{-8}$ in time with adaptive time refinement and coarsening [8] in addition to an eigenvalue-based time step restriction for ensuring numerical stability. These spatial and temporal discretizations yield numerical results that are similar to the fine resolution numerical results obtained in [32,33].

   To collect FOM snapshots, we first need to decide what time interval we utilize. To this end, in Figure 6, we plot the time evolution of the kinetic energy, $E(t)$. Figure 6 (see also Figure 1 in [32]) shows that the flow starts with a short transient interval (approximately $[0, 10]$), after which it converges to a *statistically steady state*. We emphasize that, although the flow is statistically steady, it still displays a complex, chaotic behavior. To illustrate this, in Figure 5, we display the instantaneous contour plot for the streamfunction field at $t = 40$ and $t = 60$. While $t = 40$ and $t = 60$ are well within the statistically steady state regime, the flow displays a non-periodic, complex time evolution, with a high degree of variability. Furthermore, in Figure 7, we plot the time evolution of the true ROM coefficients $a_1^{FOM}(t), a_{10}^{FOM}(t)$, and $a_{100}^{FOM}(t)$, which are obtained by projecting the FOM vorticity data onto the ROM bases, $\varphi_1$, $\varphi_{10}$, and $\varphi_{100}$, respectively:

$$a_i^{FOM}(t) = \left( \omega^{FOM}(t), \varphi_i \right), \tag{37}$$

where $\omega^{FOM}(t)$ is the FOM vorticity at time $t$. The true ROM coefficients display a non-periodic, chaotic behavior within the time interval $[10, 80]$. Thus, the numerical approximation of this statistically steady regime remains challenging for the ROMs that we investigate in this section.

   In our numerical investigation, we follow [8,32,33] and collect 701 FOM snapshots in the time interval $[T_{min}, T_{max}] = [10, 80]$ at equidistant time intervals. Collecting a large number of snapshots ensures that the FOM data used to train the ROM is rich enough to capture the relevant dynamics. Next, we use the algorithm outlined in Section 4 and the FOM snapshots to construct the ROM basis. In Figure 8, we plot selected ROM streamfunction basis functions. We observe that, as the ROM basis index increases, the spatial structures displayed by the ROM basis functions become smaller and smaller. This is consistent with the idea that the ROM modes are arranged in decreasing importance

(dominance) order: The first ROM mode is the most dominant, the second ROM mode is the second most dominant, and so on.

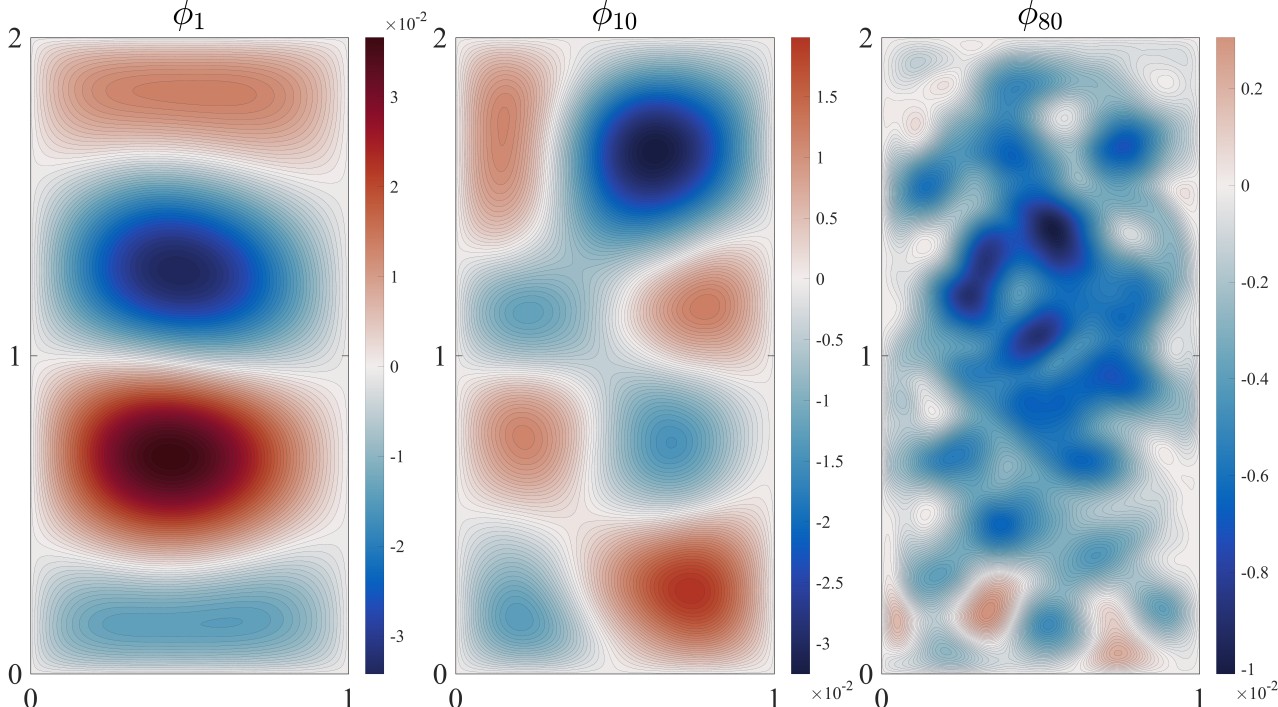

**Figure 8.** Streamfunction basis functions: $\phi_1, \phi_{10},$ and $\phi_{80}$.

### 5.5. ROM Numerical Investigation

In this section, we perform a numerical investigation of the ROM accuracy and efficiency.

To investigate the ROM accuracy, we consider the four regimes discussed in Section 5.1. First, we consider the resolved regime, both in the reconstructive (Section 5.5.1) and predictive (Section 5.5.2) settings. In these two regimes, we investigate only the standard G-ROM (25), since in the resolved case there is no need for ROM closure. The goal of these two sections is to use the two criteria presented in Section 5.3 (i.e., the relative $L^2$ error and the relative energy content) to determine the minimum ROM dimension ($r$) that is necessary in the resolved regime. Next, we consider the under-resolved regime, both in the reconstructive (Section 5.5.3) and predictive (Section 5.5.4) settings. In these two regimes, we investigate the standard G-ROM (25) and one LES-ROM, i.e., the DD-VMS-ROM presented in Section 4.2.2. The goal of these two sections is to determine whether the LES-ROM can significantly increase the standard G-ROM accuracy in the under-resolved regime.

To investigate the ROM efficiency, in Section 5.5.5 we discuss the computational cost of the standard G-ROM and the LES-ROM.

For both the G-ROM and the LES-ROM, we use the same time discretization on the time interval $[10, 80]$: the RK4 method with a uniform step size $\Delta t = 10^{-3}$.

#### 5.5.1. Resolved, Reconstructive Regime

In this section, we consider the resolved, reconstructive regime.

In Table 2, we list the relative $L^2$ errors (36) of the time-averaged streamfunction and the relative energy content (30) for G-ROM with several $r$ values: $r = 10, 20, 40,$ and 80. As expected, as the G-ROM dimension ($r$) increases, the relative errors converge to 0 and the relative energy content increases. We emphasize, however, that

one needs a relatively large $r$ value to attain what we defined as a resolved regime: *To attain an $\mathcal{O}(10^{-1})$ relative error and 90% relative energy content, one needs to take $r = \mathcal{O}(10^2)$.*

**Table 2.** Resolved, reconstructive regime. Relative $L^2$ errors (36) of the time-averaged streamfunction and relative energy content (30) for G-ROM with different $r$ values.

| $r$ | 10 | 20 | 40 | 80 | 120 |
|---|---|---|---|---|---|
| Relative error | $2.009 \times 10^2$ | $7.377 \times 10^0$ | $4.595 \times 10^{-1}$ | $2.999 \times 10^{-1}$ | $1.493 \times 10^{-1}$ |
| Relative energy content | 65.24% | 75.25% | 83.65% | 90.33% | 93.48% |

In Figure 9, for $r = 10, 40$, and 120, we plot the time-average of the streamfunction $\psi$ over the time interval $[10, 80]$ for the FOM and G-ROM. We note that we use the same scale for the FOM and the G-ROM with large $r$ values (i.e., $r = 40$ and $r = 120$). However, for the G-ROM with a low $r$ value (i.e., $r = 10$), we use a different scale, since the magnitude of these G-ROM results is much larger than the rest. The plots in Figure 9 show that the G-ROM with low $r$ values (i.e., $r = 10$ and $r = 40$) fails to recover the FOM four-gyre pattern. The G-ROM with $r = 120$ captures the FOM four-gyre pattern, but even in this case the magnitude of the time-averaged streamfunction is only marginally accurate. Thus, the plots in Figure 9 support the results in Table 2: *To recover the FOM four-gyre pattern, one needs to take $r = \mathcal{O}(10^2)$.*

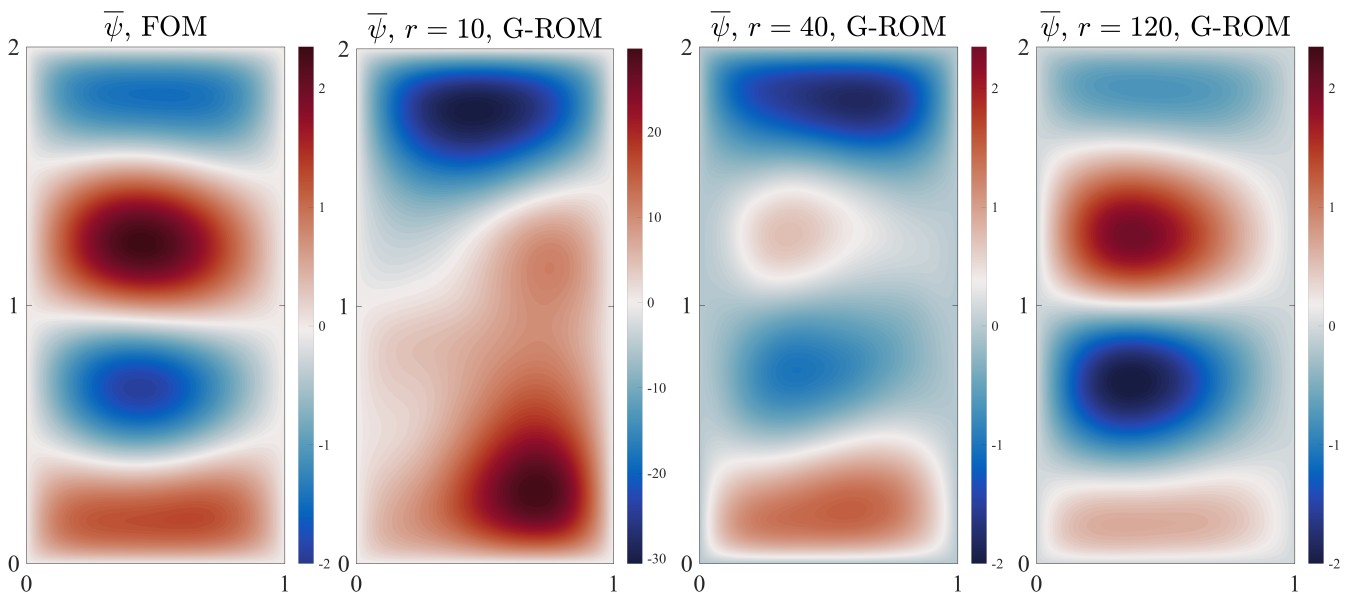

**Figure 9.** Resolved, reconstructive regime. Time-averaged streamfunction, $\overline{\psi}$, for FOM and G-ROM with $r = 10, 40$, and 120.

### 5.5.2. Resolved, Predictive Regime

In this section, we consider the resolved, predictive regime. To construct the G-ROM basis functions, we use data (snapshots) from the time interval $[10, 45]$ and test the G-ROM on a longer time interval (i.e., $[10, 80]$) to test the predictive capabilities of the G-ROM.

In Table 3, we list the relative $L^2$ errors (36) of the time-averaged streamfunction and the relative energy content (30) for G-ROM with several $r$ values: $r = 10, 20, 40$, and 80. We note that, as the G-ROM dimension ($r$) increases, the errors converge to 0 and the relative energy content increases. As expected, the errors in the predictive regime are worse than the errors in the reconstructive regime in Section 5.5.1. Furthermore, as in the reconstructive regime, one needs a relatively large $r$ value to attain what we defined as a resolved regime: *To attain an $\mathcal{O}(10^{-1})$ error and 90% relative energy content, one needs to take $r = \mathcal{O}(10^2)$.*

**Table 3.** Resolved, predictive regime. Relative $L^2$ errors (36) of the time-averaged streamfunction and relative energy content (30) for Galerkin ROM (G-ROM) with different $r$ values.

| $r$ | 10 | 20 | 40 | 80 | 120 |
|---|---|---|---|---|---|
| Relative error | $2.030 \times 10^2$ | $1.015 \times 10^1$ | $5.115 \times 10^{-1}$ | $3.892 \times 10^{-1}$ | $2.619 \times 10^{-1}$ |
| Relative energy content | 66.03% | 76.38% | 85.17% | 92.23% | 95.41% |

In Figure 10, for $r = 10, 40$, and 120, we plot the time-average of the streamfunction $\psi$ over the time interval $[10, 80]$ for the FOM and G-ROM. We note that we use the same scale for the FOM and the G-ROM with large $r$ values (i.e., $r = 40$ and $r = 120$). For the G-ROM with a low $r$ value (i.e., $r = 10$), we use a different scale, since the magnitude of these G-ROM results is much larger than the rest. The plots in Figure 10 show that, as in the reconstructive regime in Section 5.5.1, the G-ROM with low $r$ values (i.e., $r = 10$ and $r = 40$) fails to recover the FOM four-gyre pattern. The G-ROM with $r = 120$ captures the FOM four-gyre pattern, but even in this case the magnitude of the time-averaged streamfunction is only marginally accurate. Thus, the plots in Figure 10 support the results in Table 3: *To recover the FOM four-gyre pattern, one needs to take $r = \mathcal{O}(10^2)$.*

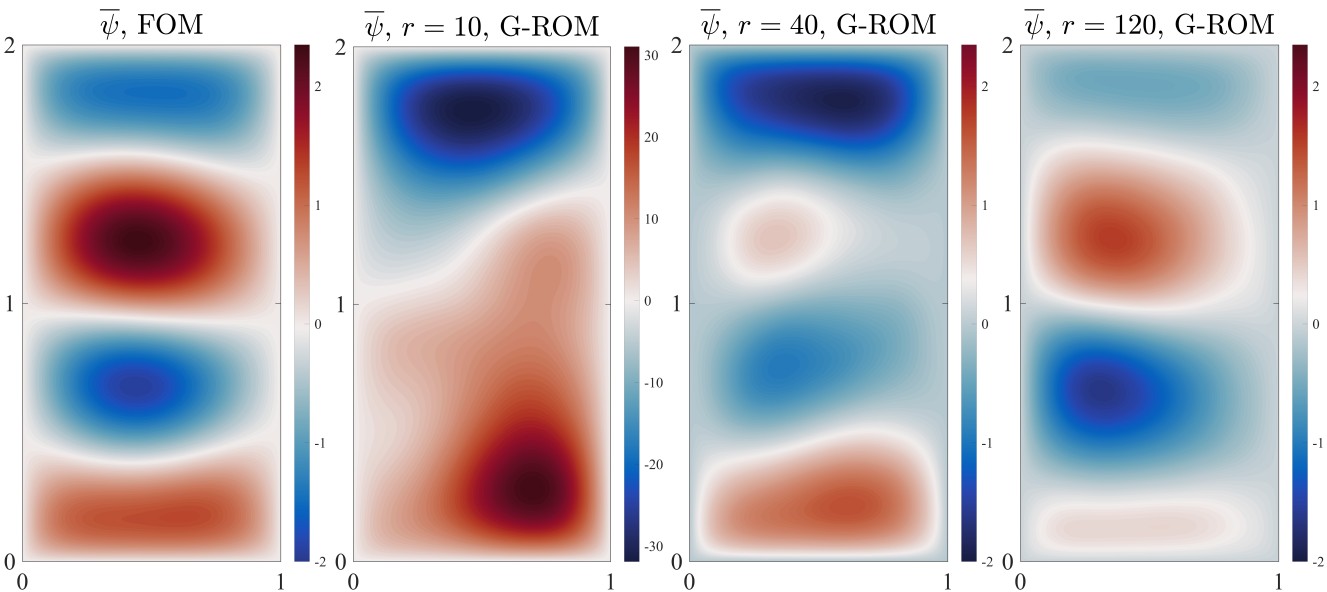

**Figure 10.** Resolved, predictive regime. Time-averaged streamfunction, $\overline{\psi}$, for FOM and G-ROM with $r = 10, 40$, and 120.

### 5.5.3. Under-Resolved, Reconstructive Regime

In this section, we consider the under-resolved, reconstructive regime. Since we use the under-resolved regime, we investigate the standard G-ROM and an LES-ROM. Specifically, we investigate the improved, three-scale version [90] of the DD-VMS-ROM (34).

In Table 4, we list the relative $L^2$ errors (36) of the time-averaged streamfunction of the G-ROM and LES-ROM for several $r$ values: $r = 10, 15$, and 20. For all the $r$ values considered, *the LES-ROM is orders of magnitude more accurate than the G-ROM.* More importantly, for $r = 20$, *the LES-ROM is almost one order of magnitude more accurate than the G-ROM with $r = 120$,* which was used in Table 2.

In Figure 11, for $r = 10$, we plot the time-average of the streamfunction $\psi$ over the time interval $[10, 80]$ for the FOM, G-ROM, and LES-ROM. We note that we use the same scale for the FOM and the LES-ROM. For the G-ROM, however, we use a different scale, since the magnitude of the G-ROM results is much larger than the rest. The plots in Figure 11 show that the G-ROM fails to recover the FOM four-gyre pattern. On the other hand, the LES-ROM successfully captures the four-gyre pattern and its correct magnitude. In fact, the LES-ROM with $r = 10$ is even more accurate than the resolved G-ROM with $r = 120$

in Figure 9. Thus, the plots in Figure 11 support the results in Table 4: *The LES-ROM is dramatically more accurate than the G-ROM.*

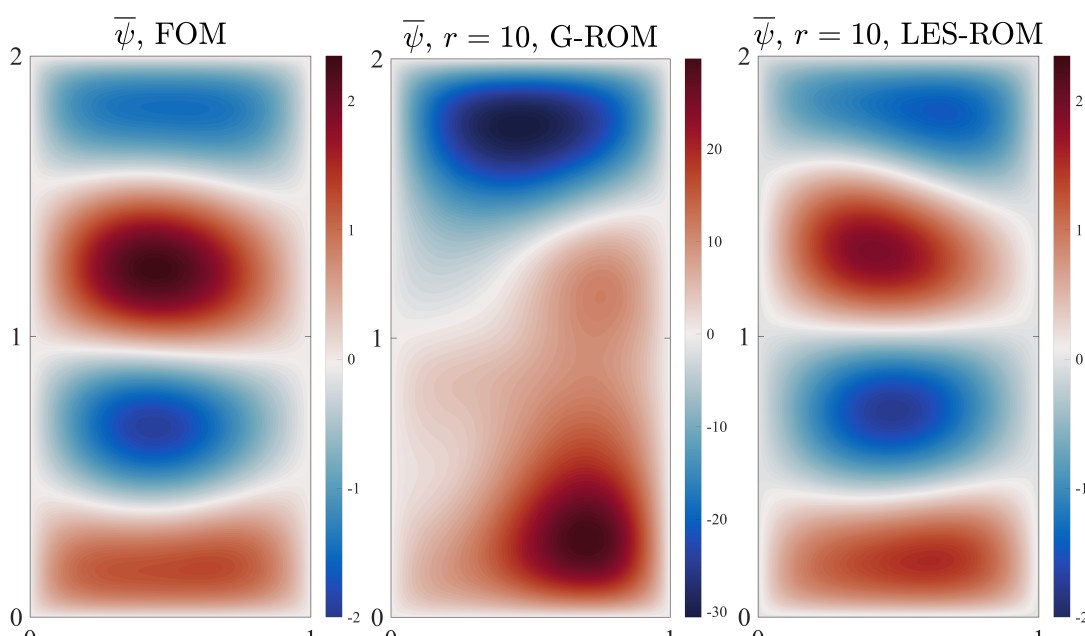

**Figure 11.** Under-resolved, reconstructive regime. Time-averaged streamfunction, $\overline{\psi}$, for FOM, G-ROM, and large eddy simulation (LES)-ROM with $r = 10$.

**Table 4.** Under-resolved, reconstructive regime. Relative $L^2$ errors (36) of the time-averaged streamfunction for G-ROM and LES-ROM for different $r$ values.

| $r$ | G-ROM | LES-ROM |
|---|---|---|
| 10 | $2.009 \times 10^2$ | $1.074 \times 10^{-1}$ |
| 15 | $5.569 \times 10^1$ | $6.780 \times 10^{-2}$ |
| 20 | $7.377 \times 10^0$ | $2.784 \times 10^{-2}$ |

5.5.4. Under-Resolved, Predictive Regime

In this section, we consider the under-resolved, predictive regime for the G-ROM and LES-ROM. To construct the G-ROM and LES-ROM basis functions, we use data (snapshots) from the time interval $[10, 45]$ and test the G-ROM and LES-ROM on a longer time interval (i.e., $[10, 80]$) to test the predictive capabilities of the G-ROM and LES-ROM.

In Table 5, we list the relative $L^2$ errors (36) of the time-averaged streamfunction of the G-ROM and LES-ROM for several $r$ values: $r = 10, 15$, and 20. For all the $r$ values considered, *the LES-ROM is orders of magnitude more accurate than the G-ROM.* Most importantly, for $r = 20$, *the LES-ROM is more accurate than the G-ROM with $r = 120$,* which was used in Table 3.

In Figure 12, for $r = 10$, we plot the time-average of the streamfunction $\psi$ over the time interval $[10, 80]$ for the FOM, G-ROM, and LES-ROM. We note that we use the same scale for the FOM and the LES-ROM. For the G-ROM, however, we use a different scale, since the magnitude of the G-ROM results is much larger than the rest. The plots in Figure 12 show that the G-ROM fails to recover the FOM four-gyre pattern. On the other hand, the LES-ROM successfully captures the four-gyre pattern and its correct magnitude. Thus, the plots in Figure 12 support the results in Table 5: *The LES-ROM is significantly more accurate than the G-ROM.*

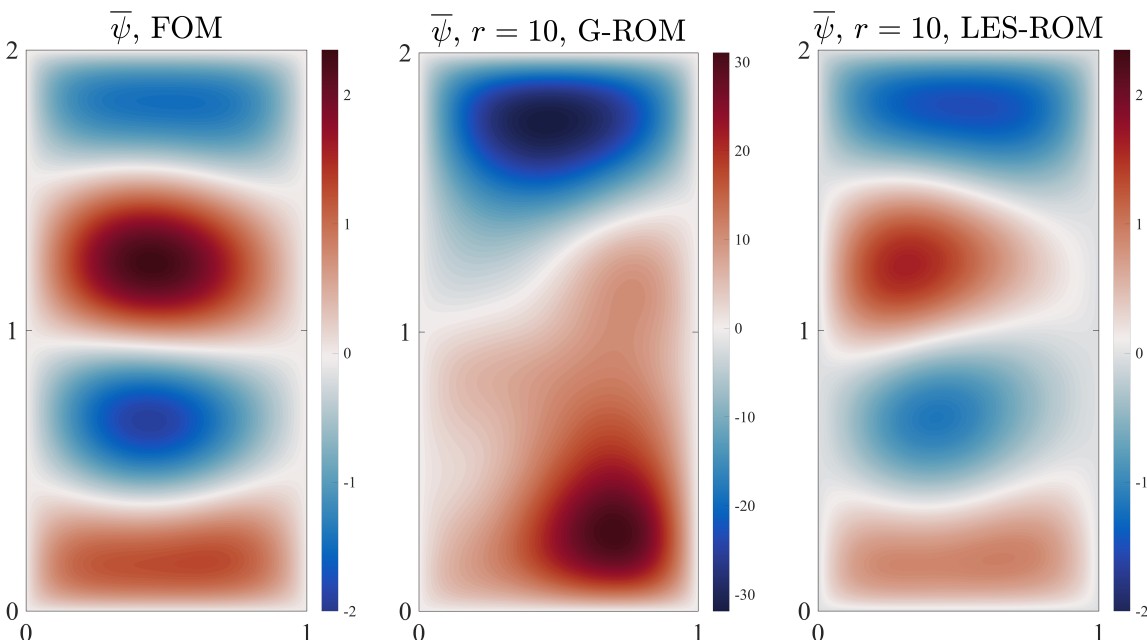

**Figure 12.** Under-resolved, predictive regime. Time-averaged streamfunction, $\overline{\psi}$, for FOM, G-ROM, and LES-ROM with $r = 10$.

**Table 5.** Under-resolved, predictive regime. Relative $L^2$ errors (36) of the time-averaged streamfunction for G-ROM and LES-ROM for different $r$ values.

| $r$ | **G-ROM** | **LES-ROM** |
|-----|-----------|-------------|
| 10 | $2.030 \times 10^2$ | $1.622 \times 10^{-1}$ |
| 15 | $2.880 \times 10^2$ | $2.385 \times 10^{-1}$ |
| 20 | $1.015 \times 10^1$ | $1.266 \times 10^{-1}$ |

### 5.5.5. Computational Cost

The ROM computational cost has two components: (i) the computational cost of the *offline stage*, i.e., when the ROM operators are assembled; and (ii) the computational cost of the *online stage*, i.e., when the ROM is actually used in practical computations. While the offline computational cost can be high, it is often offset in the online stage, when the ROM is used for numerous runs.

In Table 6, we list the CPU time for the FOM, G-ROM, and LES-ROM in the online stage. We note that the CPU time of the G-ROM is similar to the CPU time of the LES-ROM. We emphasize that *both the G-ROM and the LES-ROM CPU times are orders of magnitude lower than the FOM CPU time.* Furthermore, the G-ROM CPU time increases significantly as $r$ increases.

**Table 6.** CPU time for FOM, G-ROM, and LES-ROM in the online stage.

| FOM CPU time | | | $2.19 \times 10^5$ s | | |
|---|---|---|---|---|---|
| G-ROM CPU time | $r = 10$ $2.69 \times 10^0$ s | $r = 20$ $4.80 \times 10^0$ s | $r = 40$ $4.58 \times 10^1$ s | $r = 80$ $1.32 \times 10^2$ s | $r = 120$ $6.45 \times 10^2$ s |
| LES-ROM CPU time | | $r = 10$ $3.22 \times 10^0$ s | | $r = 15$ $3.85 \times 10^0$ s | $r = 20$ $5.07 \times 10^0$ s |

### 5.5.6. Summary

The results in our numerical investigation yield the following conclusions:

1. For our test problem, the resolved regime requires ROMs that have a *large dimension* (i.e., $r = \mathcal{O}(10^2)$) in both the reconstructive and the predictive regimes.
2. In the realistic, under-resolved regime, *the LES-ROM is orders of magnitude more accurate than the G-ROM* in both the reconstructive and the predictive regimes.
3. The LES-ROM in the under-resolved regime (i.e., with $r = 20$) is *significantly more accurate* and *dramatically more efficient* than the G-ROM in the resolved regime (i.e., with $r = 120$).

## 6. Conclusions and Outlook

The quasi-geostrophic equations (QGE) (also known as the barotropic vorticity equations) are a simplified mathematical model for large scale wind-driven ocean circulation. Since the QGE computational cost is significantly lower than the computational cost of full fledged mathematical models of ocean flows, the QGE have often been used to test new numerical methods for geophysical flows, such as reduced order models (ROMs).

In this brief survey, we summarized projection-based ROMs developed for the QGE in order to understand ROMs' potential in efficient numerical simulations of ocean flows. Specifically, in Section 2, we briefly explained how the QGE are derived from the primitive equations by using simple scaling arguments. We also outlined the various QGE formulations currently used, and we illustrated the importance of the Rossby number, which quantifies the rotation effects in the QGE. In Section 3, we surveyed the main numerical methods used in the spatial discretization of the QGE: finite difference, finite volume, pseudospectral and spectral, and finite element methods. In Section 4, we presented the main steps in the construction of the standard Galerkin ROM (G-ROM). Specifically, we showed how the full order model (FOM) simulations generate data (snapshots) that is used to build the ROM basis, which is then utilized in a Galerkin projection framework to construct the G-ROM. We also emphasized the importance of appropriate treatment of the under-resolved regime, i.e., when the number of ROM modes is not enough to capture the relevant QGE dynamics. The ROM under-resolved regime is often encountered in realistic geophysical settings dominated by convection, when the Kolmogorov n-width is large. One of the main approaches for tacking the ROM under-resolved regime is ROM closure modeling, i.e., modeling the effect of the discarded ROM modes. We reviewed two types of ROM closure models for the QGE: large eddy simulation (LES) ROM closure models (which are based on spatial filtering and data driven modeling), and machine learning (ML) ROM closure models. Finally, in Section 5, we showed how ROMs are used in the numerical simulation of the QGE. To this end, we considered a QGE test problem in which long-term time averaging yields a four-gyre pattern. We showed that, if enough ROM modes were used (i.e., in the resolved regime), the standard G-ROM yielded accurate results at a low computational cost. If, however, only a few ROM modes were used (i.e., in the under-resolved regime), the standard G-ROM yielded inaccurate results, whereas the LES-ROM yielded accurate results at a low computational cost.

ROMs have a significant potential in efficient and relatively accurate numerical simulations of geophysical flows that display recurrent dominant spatial structures. This brief survey aimed at showcasing the ROMs' potential in simplified settings, i.e., for QGE simulations. We emphasize, however, that the ultimate goal is to use ROMs in realistic many query atmospheric and oceanic applications, e.g., uncertainty quantification and data assimilation. While the first steps have been made (see, e.g., [106–112]), there are significant challenges that still need to be addressed. Next, we present several potential future research avenues in the ROM exploration of the QGE and more complex models of geophysical flows.

To develop ROMs for geophysical flows, *realistic* computational settings need to be considered. For example, realistic parameters (e.g., the Reynolds number, *Re*), and realistic complex geometries need to be investigated. Since realistic oceanic and atmospheric

flows display an enormous range of spatial and temporal scales, new ROMs need to be constructed for under-resolved regimes in which the ROM closure problem becomes central, just as in FOM. Thus, novel robust, stable, accurate, and efficient ROM closure models for realistic geophysical flows need to be built. But how should these ROM closure models be developed? By using physical insight (as in classical FOMs), data (as currently done in many research areas), or both? Furthermore, in addition to the rotation effects modeled by the QGE, *stratification* should also be investigated. In the simplified QGE setting, stratification could be included by considering the multilayer QGE or the continuously stratified QGE. More realistic western boundary layers should also be investigated. Of course, all these problems are compounded when mathematical models that are more accurate than the QGE are considered, such as the Boussinesq equations. Finally, mathematical support for these new ROMs needs to be provided. The first steps in this direction have been made (see, e.g., [61,75]), but much more remains to be done.

**Author Contributions:** Conceptualization, C.M., Z.W., D.R.W., X.X. and T.I.; methodology, C.M., Z.W., D.R.W., X.X. and T.I.; software, C.M., Z.W. and D.R.W.; validation, C.M., Z.W. and D.R.W.; formal analysis, C.M., Z.W. and D.R.W.; investigation, C.M., Z.W., D.R.W., X.X. and T.I.; resources, C.M., Z.W., D.R.W., X.X. and T.I.; data curation, C.M., Z.W. and D.R.W.; writing—original draft preparation, C.M., Z.W., D.R.W., X.X. and T.I.; writing—review and editing, C.M., Z.W., D.R.W., X.X. and T.I.; visualization, C.M., Z.W. and D.R.W.; supervision, T.I.; project administration, T.I.; funding acquisition, Z.W., D.R.W. and T.I. All authors have read and agreed to the published version of the manuscript.

**Funding:** This research was funded by National Science Foundation, DMS-2012253, DMS-1953113, DMS-1913073, OAC-1450327, and by U.S. Department of Energy, DE-SC0020270.

**Institutional Review Board Statement:** Not applicable.

**Informed Consent Statement:** Not applicable.

**Conflicts of Interest:** The authors declare no conflict of interest.

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
