# Peer review of "Reduced Order Models for the Quasi-Geostrophic Equations: A Brief Survey"

_fluids, doi:10.3390/fluids6010016_

Round 1
Reviewer 1 Report
This paper presents an overview of the Reduced Order Modeling technique applied to the quasi-geostrophic equations for large-scale geophysical flows. The paper is very well written, and covers every aspect of the ROM technique, and also touches on the basics of the modeling of large-scale geophysical flows. Overall, this is a timely survey of the latest advances with the ROM technique applied to geophysical flows. It should be indispensable reference for researchers working in these fields. I highly recommend its publication in MDPI-Fluids journal, after some minor revisions.
To the authors, I have the following questions / suggestions.
1. In (2), it is stated that a key question for ROM is to find the 'best' basis. But it doesn't seem that this question is adequately addressed later in the text. For example, in (14), it is said that minimization leads to an eigenvalue problem, which produces the basis for the ROM. But the authors did not make the connection between this problem and the question of 'best' basis. Also, please comment on the impact that the choice of snapshots could have on the question of 'best' basis.
2. Just under (7b), this is not how Ro is normally defined. You seem to suggests that the average Coriolis parameter f0 is zero here, which is not true in regions where the QGs are applicable.
3. It appears that the test in Section 5 is done without any beta effect, i.e. uniform Coriolis parameter in the y direction. With the beta effect, an intensified boundary layer should appear along the western boundary, as in Figure 1. But boundary layers were not seen in figures presented in Section 5. I would argue that a test case with the beta effect will be more appropriate for the ROM technique, because the dynamics would be highly skewed by the beta effect and a dominant and coherent structure, with an intensified western boundary current, would emerge, exactly the type of problem that ROM is most effective for. Without the beta effect, the QG is similar to the 2D Navier-Stokes equations, and as the authors clearly demonstrated, is very hard for ROM, because there is not much other than turbulence here. Hence, I would suggest the authors consider the case with the beta effect in your future studies.
4. Equation (19) is referred to before it has appeared, in the line above.
Reviewer 2 Report
The article reviews different types of reduced order methods for the Quasi-Geostrophic equations. The focus is in particular on closure models which are particularly relevant and needed in the case of the Quasi-Geostrophic equations. The article is well written and the adopted numerical methods are described accurately and in detail. The numerical results are well presented and convincing. For the above reasons, I strongly encourage the publication of the article after a minor revision. In particular, I have only a few questions/suggestions for the authors:
- Eq. 11 pag 8, a Φ should be replaced with Φh . Moreover, I would use a consistent notation to indicate the partial derivative in equation 11 and 12.
- Section 5.3, I think that a good criterion to assess the quality of the ROM, instead of the L2 error of the time-averaged field, would be the L2 error calculated both in time and space. If the calculation of this type of error is too time-consuming the authors can skip this point.
- pag 19, is there any reason why the ROM is constructed starting from t = 10 and not from the initial time instant? Is the reproduction of the build-up phase particularly challenging?
